# Climate variability can outweigh the influence of climate mean changes for extreme precipitation under global warming

Kalle Nordling[1,2], Nora L. S. Fahrenbach[3], and Bjørn H. Samset[1]

[1]Center for International Climate and Environmental Research (CICERO), Oslo, Norway
[2]Finnish Meteorological Institute, Helsinki, Finland
[3]Institute for Atmospheric and Climate Science, ETH Zurich, Zurich, Switzerland

**Correspondence:** Kalle Nordling (kalle.nordling@fmi.fi)

**Abstract.**

As global warming progresses, weather conditions like daily temperature and precipitation are changing due to changes in their means and distributions of day-to-day variability. In this study, we show that changes in variability have a stronger influence on the number of extreme precipitation days than the change in the mean state in many locations. We analyze daily precipitation and maximum temperatures at four levels of global warming and under different emission scenarios for the Northern Hemisphere (NH) summer (June – August). Our analysis is based on initial condition large ensemble simulations from three fully coupled Earth System Models (MPI-ESM1-2-LR, CanESM5, and ACCESS-ESM1-5) contributing to the Climate Model Inter-comparison Project phase 6 (CMIP6). We also use information from the Precipitation Driver Response Model Intercomparison Project (PDRMIP) to discern the influence of different climate drivers (notably aerosols and greenhouse gases). We decompose the total changes in daily NH summer precipitation and daily maximum temperature into mean and variability components (standard deviation and skewness). Our results show that in many locations, variability exerts a stronger influence than mean changes on daily precipitation. Changes in the widths and shapes of precipitation distributions are especially dominating over mean changes in Asia, the Arctic and Sub-Saharan Africa. In contrast, temperature changes are primarily driven by changes in the mean state. For the near future (2020–2040), we find that reductions in aerosol emissions would increase the likelihood of extreme summertime precipitation only over Asia. This study emphasizes the importance of incorporating daily variability changes into climate change impact assessments and advocates that future emulator and impact model development should focus on improving the representation of daily variability.

## 1 Introduction

In 2023, many regions experienced an unusually hot summer with record-breaking temperatures, widespread wildfires and heavy rainfall followed by severe flooding events (Rantanen and Laaksonen, 2024; Copernicus, 2023; wmo, 2023). Changes in climate can be driven by different natural factors, like volcanic emissions and ocean variability, as well as different anthropogenic drivers, like anthropogenic aerosol and $CO_2$ emissions. Aerosols and $CO_2$ affect regional climates differently: $CO_2$ blocks surface upwelling longwave radiation. Sulfate aerosols reflect incoming solar radiation which results in surface cooling during daytime. In contrast, absorbing aerosols, like black carbon, absorb incoming solar radiation and thus lead to a warming

of surrounding air masses (Nordling et al., 2021; Szopa et al., 2021). These different climate forcings not only affect temperatures differently but also wet and dry extremes (Sillmann et al., 2019) and the diurnal cycle (Stjern et al., 2020). However, while the effects of carbon dioxide are relatively well constrained, the impact of aerosols constitutes still one of the major uncertainties in climate science (Chen et al., 2021). For instance, while the global temperature impact of absorbing aerosols is relatively weak, they play a possibly large but still uncertain role in regional precipitation changes (Samset, 2022). The aerosol

effect on climate is further complicated by the fact that the induced climate response is dependent on the location of the aerosol emissions (Persad, 2023; Westervelt et al., 2020; Persad and Caldeira, 2018) and that the aerosol effects of locally emitted aerosols can reach far beyond their local emission regions (Wilcox et al., 2019; Fahrenbach and Bollasina, 2023). For example, Asian aerosol emissions have pronounced effects on Arctic temperatures due to changes in energy transport and albedo feedback (Merikanto et al., 2021) and the Australian monsoon due to changes in teleconnection patterns (Fahrenbach et al., 2024).

Thus, it is certainly plausible that regional aerosol emission changes induce changes in daily weather and extremes in local and remote regions.

Daily weather variability, in particular, plays a key role in extreme events and is of utmost importance when it comes to adapting to climate change since climate risk mitigation strategies depend on our understanding of day-to-day weather patterns. Changes in weather extremes are influenced by changes in the mean climate conditions (which are influenced by

global warming), variability on decadal timescales as well as day-to-day variations in weather (which are driven primarily by daily-to-annual scale internal climate variability). We have observed that extreme weather events have already changed and are continuing to do so as our planet warms (Myhre et al., 2019; Sippel et al., 2020). For example, the unprecedented summer heatwave in Europe in 2019 would have been impossible without anthropogenic climate change (Ma et al., 2020).

Previous studies have investigated changes in probability density functions (PDFs) of precipitation under global warming.

Pendergrass et al. (2017) showed that the variability of weather patterns is increasing across most regions under a warming climate. This is evident in the widening of PDFs, indicating a growing range of possible weather outcomes. Zhang et al. (2021), utilizing the HadGEM3-GC3.05 model, found that precipitation variability is increasing on all timescales, from daily variability to year-to-year differences. This study highlights that changes on short timescales are closely linked to alterations in synoptic-scale weather patterns, emphasizing the broad-reaching impacts of climate change on precipitation. Samset et al.

(2019b) focused on the evolution of regional PDFs under global warming, particularly focusing on changes in daily PDFs of temperature and precipitation. Using the CESM1 large ensemble, they discovered that even a modest increase in global temperature (+1.5°C) results in significantly more variable precipitation over regions like Africa and South America. Katzenberger et al. (2022) studied the future precipitation variability over the Indian monsoon region and found that the likelihood of extreme rainfall is expected to increase significantly (up to sixfold) by the end of this century depending on future emissions.

This illustrates the severe regional impacts of climate change, particularly in areas which are already vulnerable to extreme weather events.

When it comes to temperature, there is a clear footprint of global warming on the change in temperature variability. In high latitudes, the annual temperature variability tends to decrease, whereas it increases in lower latitudes in the near future (Kotz et al., 2021). However, this pattern varies between seasons and models. Suarez-Gutierrez et al. (2020) investigated how

temperature-related extreme events evolve with global warming using the MPI-GE large ensemble. They discovered that daily temperatures exceeding 50°C become more common in the Arabian Peninsula, northern India, and Pakistan at a global warming level of 2°C. However, beyond the 2°C threshold, these extreme temperatures are expected to occur on every continent. Future emissions play an important role in shaping how variability and extreme weather events will change in the near- to far future. For example, Wilcox et al. (2020a) shows that a reduction of aerosol emissions in the near future could lead to an increase in the Asian summer monsoon. Understanding these dynamic changes is crucial when evaluating future extreme changes on a regional scale.

What remains unclear is the role of variability: Are precipitation and temperature extremes becoming more severe due to changes in the mean state, or due to changes in day-to-day variability? Another uncertainty relates to the climate models themselves. Despite generally agreeing on the direction of changes in extreme precipitation, the current state-of-the-art climate models show significant uncertainty regarding the magnitude of these changes, especially at regional scales. In particular, the different implementations for anthropogenic aerosols and different climate sensitivities of different ESMs add to this uncertainty. Another gap in the current knowledge is how to translate the changes in the daily distribution of weather variability to meaningful quantities, like the number of extreme weather events.

In this study, we focus on examining how daily variability in the Northern Hemisphere (NH) summer precipitation and daily maximum temperature is evolving under global warming and different emission scenarios. We also show result for NH winter months in the appendix. Using large ensemble simulation from CMIP6, we investigate changes in the mean and variability (characterized by the width and shape of the PDFs) using a similar method as in van der Wiel and Bintanja (2021); Samset et al. (2019b); Lund et al. (2023) and further identify the key anthropogenic drivers (aerosols or greenhouse gases) of those changes. Our results show key regions where changes in extremes are driven by changes in variability rather than the mean state. By examining the daily variability of weather in the context of a changing climate, we can improve our understanding of the challenges and opportunities for climate change adaptation.

## 2 Method and data

### 2.1 Analysis of changes in mean and variability using PDFs

We are using simulations from single-model initial-condition large ensembles (SMILEs) from CMIP6, similarly as (Samset et al., 2019b) who studied how daily weather at a regional scale changes with global warming. In SMILEs each model is run multiple times with the same forcing and model configuration but different initial states. Figure 1 illustrates our methodology of defining daily PDFs for precipitation and maximum temperature using daily CMIP6 data from CanESM5, MPI-EMS1-2 and ACCESS-ESM-1-5. First, we defined the 1–4°C Global Warming Levels (GWLs) following the definition outlined in the IPCC AR6 report (Lee et al., 2021) (see Figure 1a). For this, a 20-year centered running mean of annual temperature for each ensemble member is calculated and the GWL is then defined as the period ± 10 years from the first year in which the global warming threshold was surpassed. A PDF is defined in this way for each grid point, which can be used to find changes in both mean and variability (here referred to as the "PDF of total change"). The second step involves removing the annual cycle at each

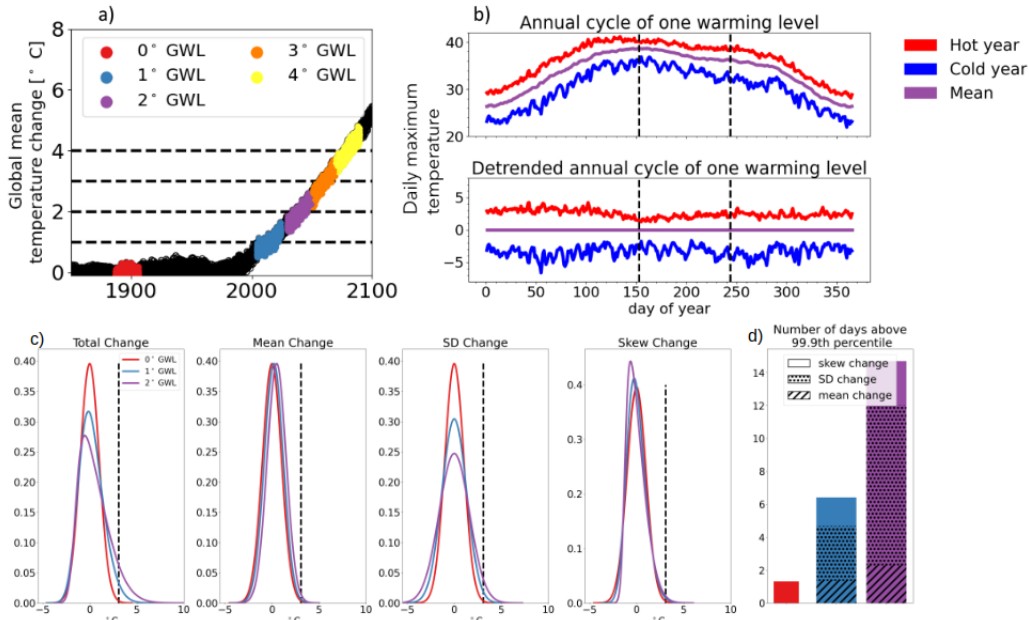

**Figure 1.** Method description. a) Selection of years for each global warming level. b) Example of regional daily maximum temperature without and with subtracting off the annual cycle to extract daily variability. c) Idealized PDFs of the total changes, the decomposition in mean, standard deviation and skewness changes as well as changes in the number of extreme days.

grid point for each GWL which gives a PDF that only differs in daily variability (shape and width of the PDF) for each GWL. These results in PDFs for each GWL which differ only by the influence of change in standard deviation, kurtosis and skewness. We quantify changes to the daily mean by calculating the difference between the means of the GWL and Pre-Industrial (PI) PDFs and then shifting the PI (0 GWL) PDF by the corresponding amount. Figure 1c, illustrates the PDF changes due to the 1) total change, 2) change in the mean and 3) change in variability (standard deviation (SD) and skewness). The final step is to calculate the number of days during which extreme weather events occur for each PDF (see Figure 1d). Here, an extreme event is defined as one that exceeds the 0.999th quantile. The return period for these extremes, as simulated by the different models, is approximately 10 years. Thus, the extreme events analyzed in this paper refer to events occurring once or less every ten years in the pre-industrial era.

For the near-future analysis, we follow the same process described above to define the PDFs but calculate the PDFs for four different Shared Socioeconomic Pathway (SSP) scenarios (SSP1-2.6, SSP2-4.5, SSP3-7.0, and SSP5-8.5; O'Neill et al., 2016) over three distinct time periods (2025–2034, 2035–2044, and 2045–2054) instead of using GWLs. For each time period and each SSP scenario, there is an underlying PDF, which we refer to as the "PDF of total change", similar to the GWL analysis. We then remove the annual cycle as in the GWL analysis to obtain a PDF that contains only changes attributable to variability (change in SD and skewness). Here, we quantify changes to the mean by calculating the difference between the means of the given SSP (and time period) and the PI PDF, and then shifting the PI (0 GWL) PDF by the corresponding amount.

These time periods are chosen to represent the largest differences in aerosol pathways across the different SSPs where the full range of uncertainty in greenhouse gas emissions has not yet emerged (although they are not negligible and are included in our simulated climate response) (Lund et al., 2019a; Wilcox et al., 2020b; Guo et al., 2021). SSP1-2.6 includes a rapid reduction in global aerosol emissions until 2050, except for an increase over southern Africa due to rapid industrialization. The aerosol emissions in SSP2-4.5 and SSP5-8.5 show a similar, but weaker, pattern, with a decrease over the NH and increase in the Southern Hemisphere (SH) as well as a strong Asian aerosol dipole (i.e., a large increase over South Asia and large decrease over East Asia) until the 2040s (Wilcox et al., 2020b; Samset et al., 2019a). The main difference between SSP2-4.5 and SSP5-8.5 lies in the black carbon (BC) emissions from South Asia which show an increase and decrease until the 2040s, respectively, as well as the aerosol emissions over South America related to different rates of deforestation (Lawrence et al., 2016). SSP3-7.0 also shows an NH decrease and SH increase in emissions. However, the sulfur dioxide (precursor of sulfate aerosols) emissions stay nearly constant over East Asia but increase over South Asia, with opposite changes in BC emissions (Wilcox et al., 2020b). The comparison of climate responses under SSP1-2.6 and SSP3-7.0, thus, allows us to investigate the influence of anthropogenic aerosols on the PDF changes, as greenhouse gas emissions remain relatively constant in these SSPs and only aerosol emissions are decreasing in SSP1-2.6. We can estimate the effects of aerosols by comparing SSP1-2.6 with SSP3-7.0, as the most significant aerosol reductions occur in Southeast and South Asia under SSP1-2.6 (Lund et al., 2019b).

## 2.2 Data

### 2.2.1 CMIP6 data

We utilize large-ensemble simulations for the SSP1-2.6, SSP2-4.5, SSP3-7.0, and SSP5-8.5 scenarios performed by three CMIP6 models, namely MPI-ESM1-2-LR (Mauritsen et al., 2019), CanESM5 (Swart et al., 2019) and ACCESS-ESM5-1.5 (Ziehn et al., 2020). Table 1 gives the model resolutions and number of ensemble members for each model. We use the same models as Lund et al. (2023) for which the summertime variability of precipitation and daily maximum temperature was verified using ERA-5 data.

**Table 1.** List of the CMIP6 Large ensemble models used in this study which performed the required SSP1-2.6, SSP2-4.5, SSP3-7.0 and SSP5-8.5 simulations. The equilibrium climate sensitivity values are taken from Zelinka et al. (2020).

| Model | Ensembles | Horizontal resolution | ECS value | Aerosol forcing | Reference |
|---|---|---|---|---|---|
| ACCESS-ESM-1-5 | 29 | $1.9° \times 1.3°$ | 3.88 | Interactive | Ziehn et al. (2020) |
| CanESM5 | 23 | $2.8° \times 2.8°$ | 5.64 | Interactive | Swart et al. (2019) |
| MPI-ESM1-2-LR | 11 | $1.9° \times 1.9°$ | 3.03 | MACv2-SP | Mauritsen et al. (2019) |

### 2.2.2 PDRMIP data

We also use idealized single forcing simulations from the Precipitation Driver Response Model Intercomparison Project (PDR-MIP; (Myhre et al., 2017)) to assess the expected impacts of different anthropogenic drivers on daily weather variability. In

**Table 2.** List of models which participated in PDRMIP and performed the coupled global experiments (CO2×2, SUL×5, BC×10).

| Model | Horizontal resolution | Aerosol setting | reference |
|---|---|---|---|
| CanESM2 | 2.8°×2.8° | Emissions | Arora et al. (2011) |
| CESM1-CAM4 | 2.5°×1.9° | Fixed concentrations | Gent et al. (2011) |
| CESM1-CAM5 | 2.5°×1.9° | Emissions | Hurrell et al. (2013); Otto-Bliesner et al. (2016) |
| GISS-E2-R | 2.0°×2.5° | Fixed concentrations | Schmidt et al. (2014) |
| HadGEM2-ES | 1.9°×1.3° | Emissions | Collins et al. (2011); Martin et al. (2011) |
| HadGEM3-GA4 | 1.9°×1.3° | Fixed concentrations | Bellouin et al. (2011b); Walters et al. (2014) |
| IPSL-CM5A | 3.8°×1.9° | Fixed concentration | Dufresne et al. (2013) |
| NorESM1-M | 2.5°×1.9° | Fixed concentrations | Bentsen et al. (2013); Kirkevåg et al. (2013); Iversen et al. (2013) |
| MIROC-SPRINTARS | 1.4°×1.4° | HTAP Emissions | Takemura et al. (2009, 2005); Watanabe et al. (2010) |

particular, we focus on experiments simulating a global doubling of $CO_2$ concentrations (hereafter CO2×2), a five-fold in-
crease in sulfate concentrations or emissions (hereafter SUL×5) and a ten-fold increase in black carbon concentrations or
emissions (hereafter BC×10) relative to the year 2000. We use the multi-model mean across nine CMIP5-generation models
which participated in PDRMIP to get a robust estimate of daily variability changes (Table 2). Throughout the analysis, we
examine the years 50–100 of the coupled simulations, discarding the first decades as spin-up. For the extreme event definition
for PDRMIP, we use the 0.90 percentile threshold to ensure that enough data is available to accurately estimate variability. The
different definition of an extreme event compared to CMIP6 analysis described above is due to the fact that PDRMIP consists
of only one member ensemble per model.

## 3  Results

### 3.1  Expected change in daily variability due to different anthropogenic drivers

Here, we first examine changes in daily weather variability in response to global increases in $CO_2$, sulfate and black carbon
aerosols simulated as part of PDRMIP. Figure 2 shows how a five-fold global increase in sulfate emissions (first column),
tenfold increase in black carbon emissions (second column) and doubling of $CO_2$ concentrations (third column) affect the
number of days of precipitation above the 90[th] percentile in preindustrial conditions. In $CO_2$×2, all models show an increase
in intense summertime precipitation over Asia, although the exact pattern over Asia differs between models (Figure A1).
NorESM1 shows the smallest changes in intense precipitation overall, with strong increases being located around the Tibetan
plateau. These changes correlate with changes in the SD. Other common features among the models include a decreasing
number of intense precipitation over the southern part of Europe (Fig 2). Spatial correlation values between changes in SD and
changes in number of days of extreme vary from 0.22 to 0.49 (Figure A4), indicating that changes in the SD can explain some
of the changes in extremes but not all.

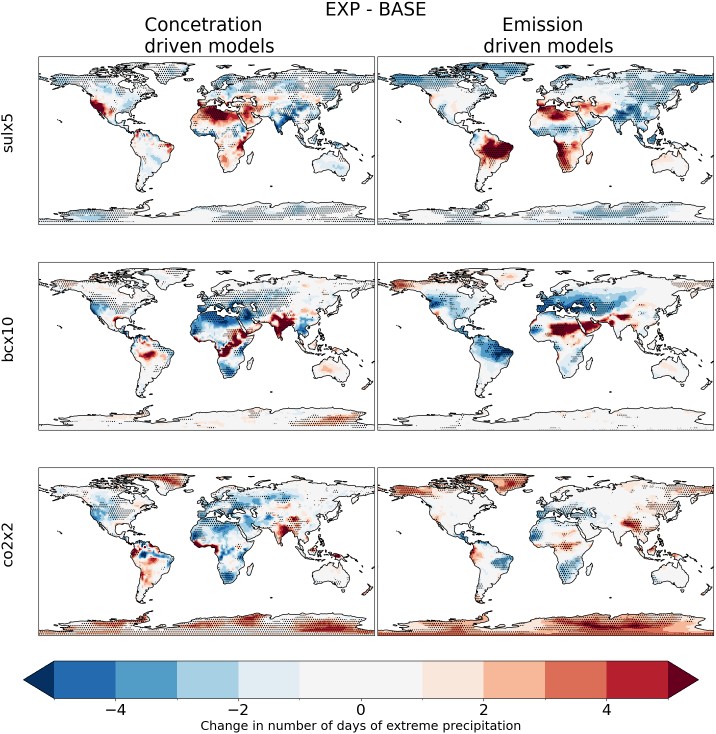

**Figure 2.** Changes in the average number of days per year of extreme (0.90 quantile) precipitation due to global doubling of $CO_2$ concentrations (CO×2), a five-fold increase in sulfate emissions (SUL×5) and a ten-fold increase in black carbon (BC×10) emissions as simulated by PDRMIP models. Left column shows the results for concentration-driven models and right column for emission-driven models. Stippling indicates where all emission- or concentration-driven models agree on the sign of change.

The impact of aerosols differs from those of $CO_2$. The climate response in SUL×5 shows a similar pattern but of opposite sign to those of $CO_2$, as expected since sulfate aerosols cool the climate while greenhouse gases warm it. For instance, HadGEM3 shows a decrease in precipitation extremes over Asia, Sub-Saharan Africa and an increase over Europe, with all these signals being opposite to the response in the doubling of $CO_2$. Additionally, all models simulate a decrease in the number of extreme days across the high latitudes in the NH. The spatial correlation between the SD and the number of extreme days for SUL×5 varies from 0.42 to 0.61 (Figure **??**). These SD differences are significant at a p-level < 0.05 using the Kolmogorov–Smirnov test.

For BC×10, the spatial correlation between changes in SD and changes in the number of extreme days is quite variable and ranges from 0.44 to 0.74 (Figure A6). The results show a higher correlation between changes in SD and extremes for the aerosol simulations than for the CO2×2 experiment. This indicates that aerosols lead to a wider/narrower distribution and thus more days of extreme precipitation than the influence of $CO_2$. Additionally, the effect of aerosols is highly regionally dependent whereas the PDFs to a $CO_2$ increase are getting wider over all regions.

## 3.2 Changes in extreme events under global warming

Changes in the probability of extreme precipitation events (>99[th] percentile) due to global warming can be attributed to two primary factors: changes in the mean state and variability. The combined impact of these two contributing factors is depicted in Figure 3, which illustrates how extreme precipitation events are evolving in response to global warming. The spatial pattern of these changes in extreme precipitation closely follows the overall pattern of annual precipitation changes, as discussed in (IPCC, 2021). In essence, regions that were already dry are experiencing increased in dryness, while areas with high climatological levels of precipitation are becoming even wetter (Feng and Zhang, 2015; Xiong et al., 2022). To provide a more detailed understanding of the total changes highlighted in Figure 3, one can decompose these changes into two components: changes in variability (as shown in Figure 4) and changes in the mean state (as shown in Figure 5).

Figure 4 shows how changes in precipitation variability are changing the likelihood of extreme precipitation events, defined as those events that occur more than once every decade in the pre-industrial era. This phenomenon is observed globally, with an overall increase in the number of such extreme events in most regions. However, there are notable exceptions: In regions like the Amazon basin, Southern Africa, and Australia, there is a slight decrease in extreme precipitation events during the NH summer months. Already, a one-degree change in global warming shows a significant increase in the likelihood of extreme precipitation, especially over the Sahel region, as simulated by MPI-ESM1-2-LR and CanESM5.

While there is broad agreement among the models about the increase in extreme precipitation, particularly in Asia, there are differences in the exact location of these changes. The most significant changes in extreme summer precipitation due to variability are seen in three main regions: South East Asia and South Asia, Sub-Saharan Africa, and the Arctic region. Each of these areas shows a distinct pattern in the increase of extreme precipitation events, underscoring the diverse impacts of changing precipitation variability across different parts of the world.

Changes in precipitation patterns can also be influenced by shifts in the mean state of precipitation driven by global warming. Figure 5 provides an overview of how shifts in the mean state affect the number of extreme precipitation days, although those changes are not as large as those resulting from shifts in variability. A consistent increase in dry days can be seen over southern Europe and, to a large extent, North America (Figure 5). Figure 6 shows whether changes in the mean state (shown in brown) or variability (shown in purple) are the dominant factors influencing the overall change in extreme precipitation events. The relative importance of change in the mean state and change in variability is defined by $\frac{\Delta variablity - \Delta Mean}{\Delta variablity + \Delta Mean}$. All three models agree on the spatial pattern of changes in variability. In particular, changes in the mean state are the dominant driver of changes in extreme precipitation events over South America, Southern Africa, and Australia. Conversely, changes in variability play a more pronounced influence on extreme precipitation changes over Eurasia.

The behaviour of daily maximum temperature during summer is quite different from that of daily precipitation (Figure 6). While changes in daily precipitation are primarily driven by changes in variability, daily maximum temperatures are predominantly influenced by changes in their mean state. On a global scale, all regions experience an increase in extreme daily maximum temperatures due to shifts in the average daily maximum temperature. In a four-degree warmer world, the daily maximum temperature distributions in almost every region are shifted outside pre-industrial ranges. However, there is some

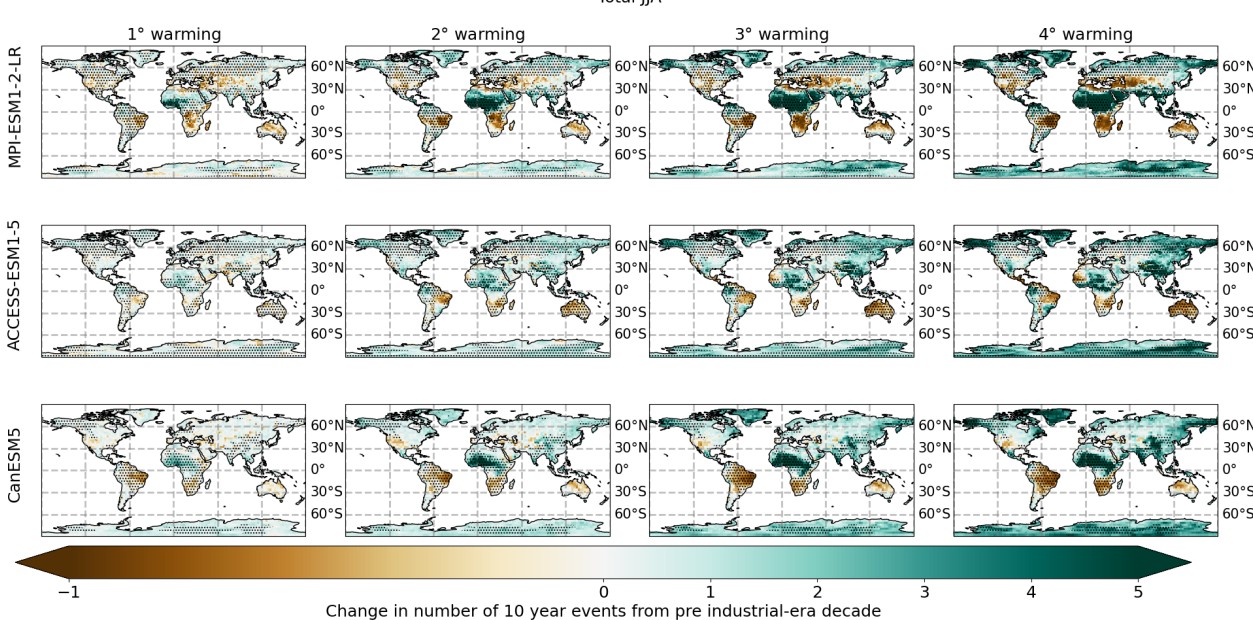

**Figure 3.** Total change in the number of days of intense precipitation events during JJA under different global warming levels. Stippling indicates regions where changes in PDFs are significant at p > 0.05

variability among climate models regarding spatial patterns of the increase in daily maximum temperatures. For instance, only the ACCESS-ESM1-5 model predicts that all summer days in Alaska will surpass rarely observed (0.999 quantile) pre-industrial temperatures in a four-degree warmer world (see Figure A7).

### 3.3 Different climate drivers in the near future

In the near future, the Earth's climate will be influenced by different anthropogenic drivers depending on different future emission scenarios and associated emission reductions. Above, we have shown the influence of different anthropogenic drivers on Earth's climate using idealized PDRMIP simulations (Section 3.1). It is not evident that different anthropogenic drivers have an effect on rare extreme events that occurred only once per decade during the pre-industrial period in the near future SSP scenarios. However, when examining more frequent extreme events (events which occur once per year), differences between aerosol-driven changes and greenhouse gas-induced warming become evident. Figures C1, C2 and C3 show changes in the likelihood of these extremes in the near future under different SSP scenarios (particularly, SSP1-2.6, SSP2.4-5, SSP3-7.0, and SSP5-8.5) for all three models due to changes in variability. Similar to the changes in extremes under global warming, the most distinct near-future changes are seen in Sub-Saharan Africa, where greenhouse gas emissions are expected to dominate. In contrast, most of the reduction in aerosol emissions is expected to occur over Asia in the future (Lund et al., 2019b).

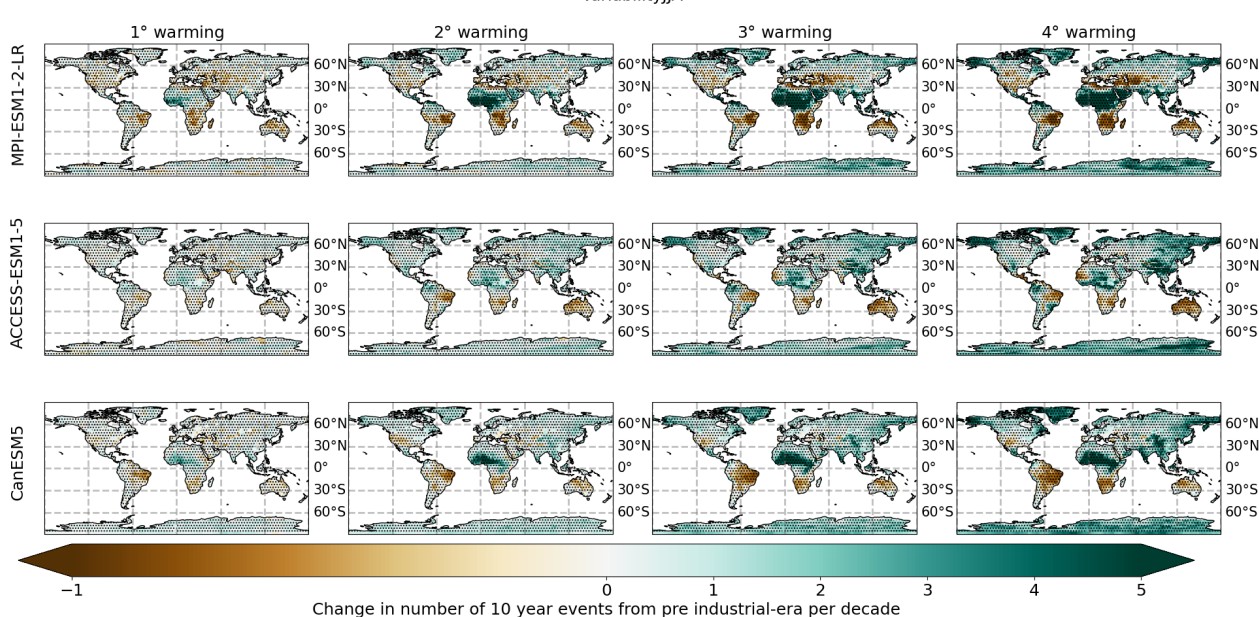

**Figure 4.** Changes in the number of days of intense JJA precipitation events due to changes in variability under different global warming levels. Stippling indicates regions where changes in PDFs are significant at p > 0.05

We can estimate the effects of aerosols by subtracting the changes seen in SSP3-7.0 from SSP1-2.6, where the most drastic aerosol reductions occur over South East and South Asia (Lund et al., 2019b). While greenhouse gas emission and land use changes will also contribute, previous work has found this method to give a reasonable first approximation of the aerosol influence over the coming decades (Wilcox et al., 2020a). Figure 7 shows the effect of aerosol emission reductions according to the SSP1-2.6 scenario for the three different climate models over Asia (for the global pattern see Figure C4). There is no model agreement on the pattern or sign of change over most land regions. The CanESM5 model suggests that the increase in the likelihood of extreme precipitation events is continuously reduced in the near future in South and East Asia regions with a continuous reduction in aerosol emissions. In contrast, MPI-ESM1-2-LR indicates a slight decrease in extreme weather events from 2025 to 2034, followed by an increase from 2035 to 2044 over the Tibetian Plateau. This would indicate that reducing aerosol emissions might make extreme weather more likely during this latter period. The ACCESS-ESM1-5 model shows the most prominent effect: A reduction of aerosol emissions leads to a clear rise in the chance of extreme rain or snow events between 2035 and 2044. This seems aligned with previous results which showed that anthropogenic aerosols suppress precipitation, including extreme precipitation, over Asia (Yang et al., 2022; Wilcox et al., 2020a; Persad, 2023).

These model differences likely stem from differences in the implementation of aerosols as well as the model's sensitivity to greenhouse gases. MPI-ESM1-2-LR uses a simplified approach, namely the MAC-SPv2 parametrization (Stevens et al., 2017), to represent aerosols (black carbon and sulfate) which only accounts for the first indirect aerosol effect without considering

change in mean JJA

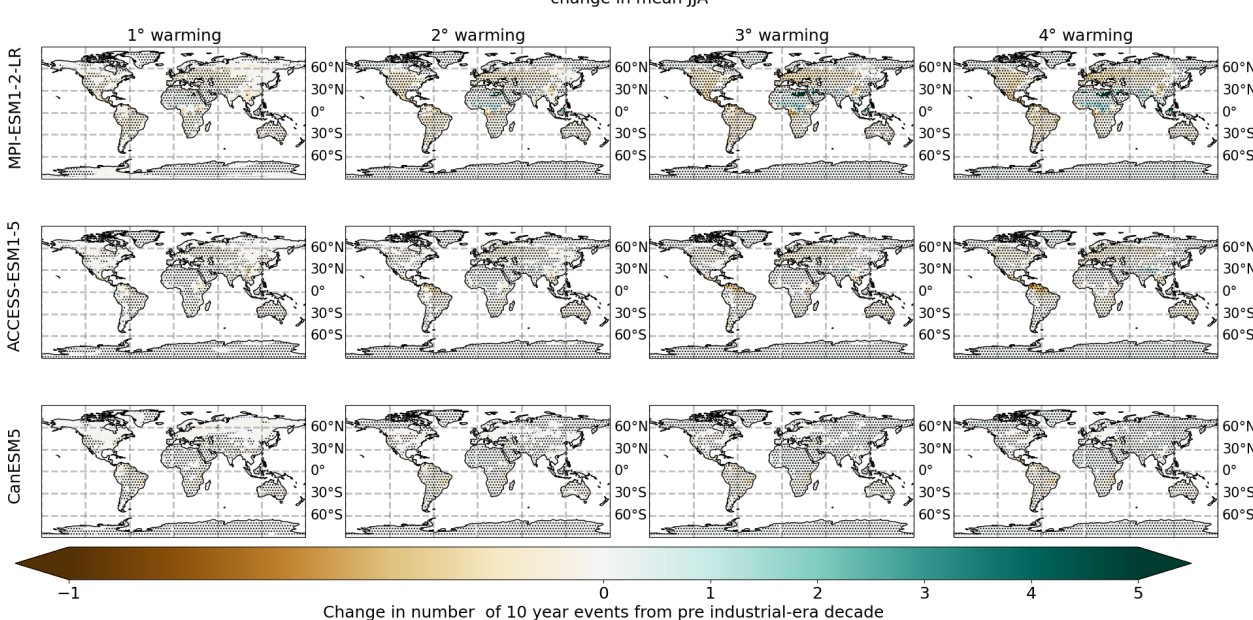

**Figure 5.** Changes in the number of extreme JJA precipitation events due to changes in the mean under different global warming levels. Stippling indicates regions where changes in PDFs are significant at p > 0.05

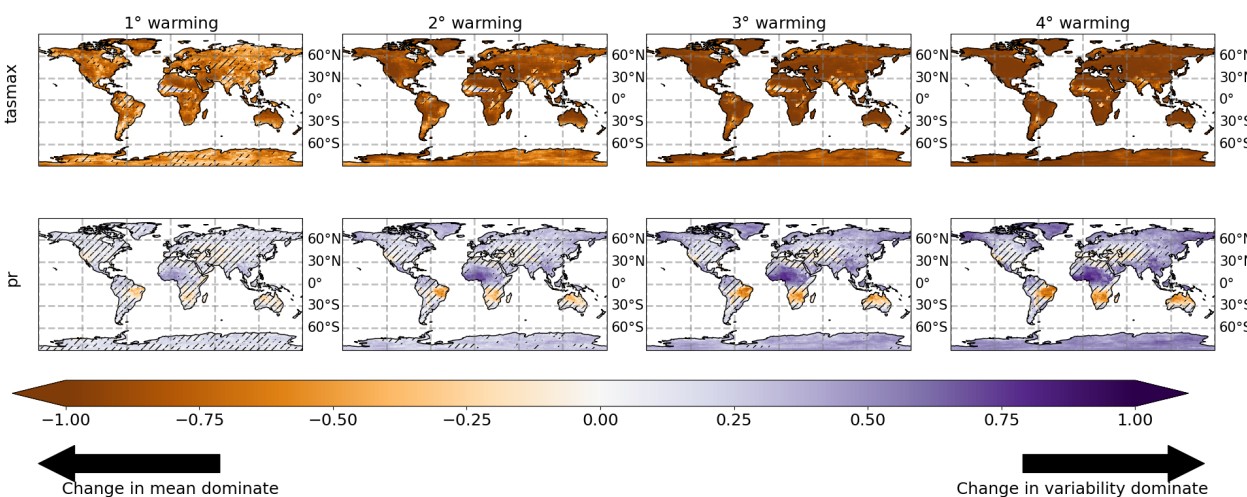

**Figure 6.** Decomposition of regional changes in JJA precipitation and daily maximum temperature extremes into changes in the mean and changes in variability under four warming levels (columns). Figure shows mean of three models and hatching indicates regions where all three models do not agree. Orange colors indicate regions where the change in the mean dominate changes in the extremes and purple colors indicate regions where variability dominates.

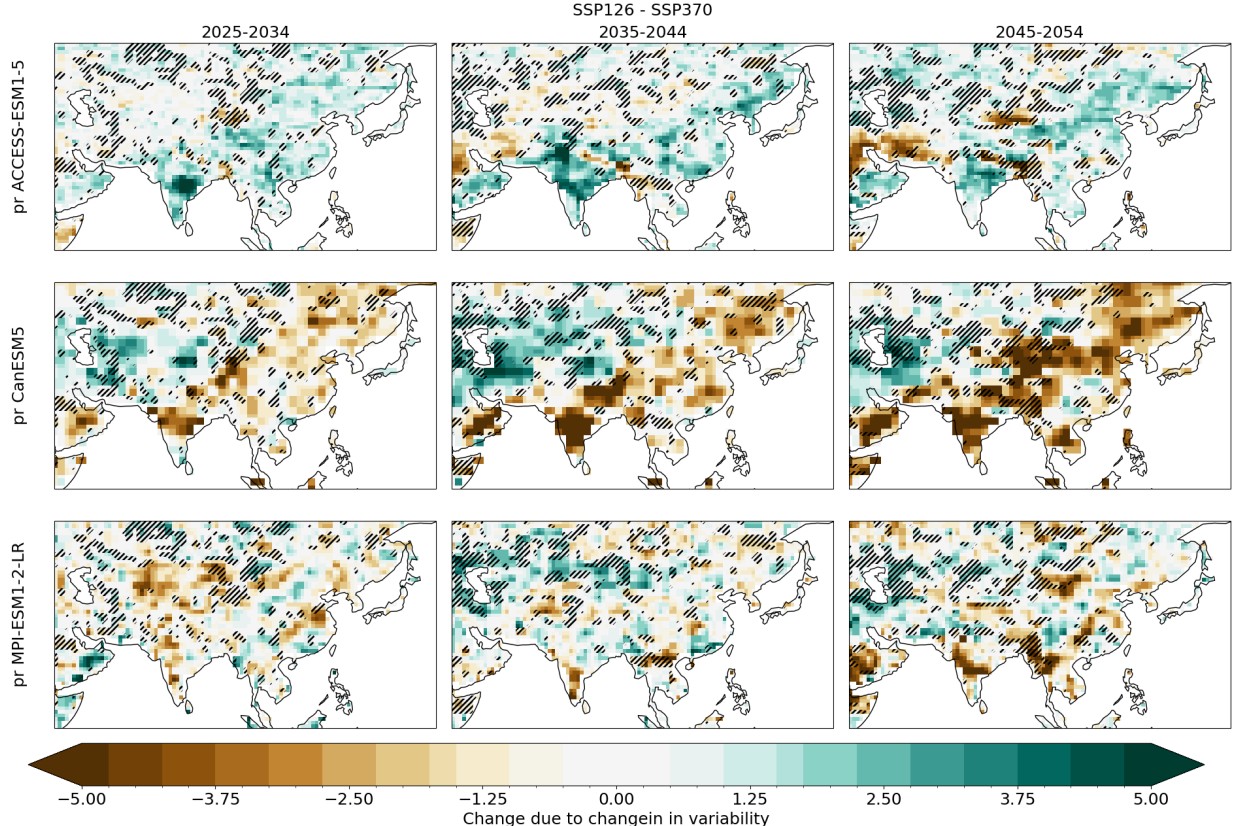

**Figure 7.** Change in the likelihood of days of extreme JJA precipitation between SSP1-2.6 and SSP3-7.0 over Asia for three different models ACCESS-ESM1-5 (row 1), CanESM5 (row 2) and MPI-ESM1-2-LR (row 2). Hatching indicates regions where all three models agree on the sign of the change.

more complex interactions. CanESM5 has a very high climate sensitivity (see Table 1), leading to greenhouse gas-dominated responses even when aerosol emissions are reduced. CanESM5, further, has a high atmospheric absorption value due to black carbon which is likely masking part of the cooling effect due to sulfate aerosols (Fiedler et al., 2023). ACCESS-ESM1-5, on the other hand, employs the CLASSIC aerosol model (Ziehn et al., 2020; Bellouin et al., 2011a; Mackallah et al., 2022), which is a very detailed representation considering seven different aerosol types and including direct and indirect effects.

## 3.4 Model discrepancies

While all models used here show similar regional changes in the likelihood of summertime extreme precipitation, they have different underlying PDFs and associated impacts on the likelihood of extremes. Figure 8 shows regional mean PDFs for total changes in daily summertime precipitation for South Asia (SAS), West Africa (WAF) and North West North America (NWN)

(using the region definitions from the IPCC report (Iturbide et al., 2020)). All these regions show a significant increase in the number of intense precipitation days due to changes in variability.

For SAS, the underlying PDFs are quite different between the individual models. The most prominent difference relates to changes in the kurtosis (Figure D1). ACCESS-ESM1-5 and CanESM5 show higher kurtosis values than MPI-ESM1-2-LR. CanESM5 is the only model that shows decreasing kurtosis with global warming. Nonetheless, all models show a similar widening of the distributions with global warming.

Over the WAF region, all three models exhibit a similar evolution in skewness. With global warming, the MPI-ESM1-2-LR and CanESM5 distributions are getting wider, indicating an increase in daily variability and an associated increase in extremes at both ends. CanESM5's evolution in standard deviation plateaus after two degrees of global warming. While CanESM5 shows a widening of the PDF, similar to the other two models, it also shows a clear change in the mean of the distribution. As a result, the likelihood of extreme values primarily increases at the high end of the tails. The most prominent discrepancy between the models is in the evolution of kurtosis, where it shows an increasing trend in MPI-ESM1-2-LR and ACCESS-ESM1-5 but decreasing trend in CanESM5.

Over NWN, all three models exhibit similar PDF shapes (while the distributions are statistically different). However, the model responses diverge regarding the PDF evolution under global warming. CanESM5 shows a change in the mean and little change in the width, whereas ACCESS-ESM1-5 and MPI-ESM1-2-LR changes are mostly in width and shape. Despite these discrepancies in underlying PDFs, all three models show a robust increase in summertime variability under global warming, which leads to an increased likelihood of extreme precipitation in the Arctic, Asia and Africa.

The next question is which change dominates the overall changes: Does the change in the SD or the skewness dominate? Figure 9 shows how these two measures change in the three different regions (WAF, NWN and SAS) and for the different models, and how this relates to change in the likelihood of extreme days. Each marker in the figure 9 represents one grid point. For CanESM5 most of the changes are due to changes in the skewness (shape of the PDF) and the underlying PDFs are even getting narrower. Both ACCESS-ESM1-5 and MPI-ESM1-2-LR show an increase of SD together with an increase in skewness.

Another interesting aspect is the regional dependence of the relative roles of changes in SD versus skewness. Here, we find that each region and model behaves differently. Over WAF, MPI shows large changes in both skewness and SD, whereas CanESM5 shows small changes in SD over WAF. Interestingly, CanESM5 shows the largest change in SD over SAS. These findings highlight that each region responds differently to global warming and that there is significant model uncertainty regarding how variability changes.

## 4 Discussion

What physical mechanisms drive the changes in variability, and what is the relationship between different mechanisms and changes in SD and skewness? We calculated power spectral densities for each region for all three models to evaluate the dependence on different timescales (Figure D2). Zhang et al. (2021) performed a moisture budget analysis on a Parameter Perturbed Ensemble of the HadGEM3-GC3.05 model. Compared to initial condition ensembles, this also samples the uncertainty from

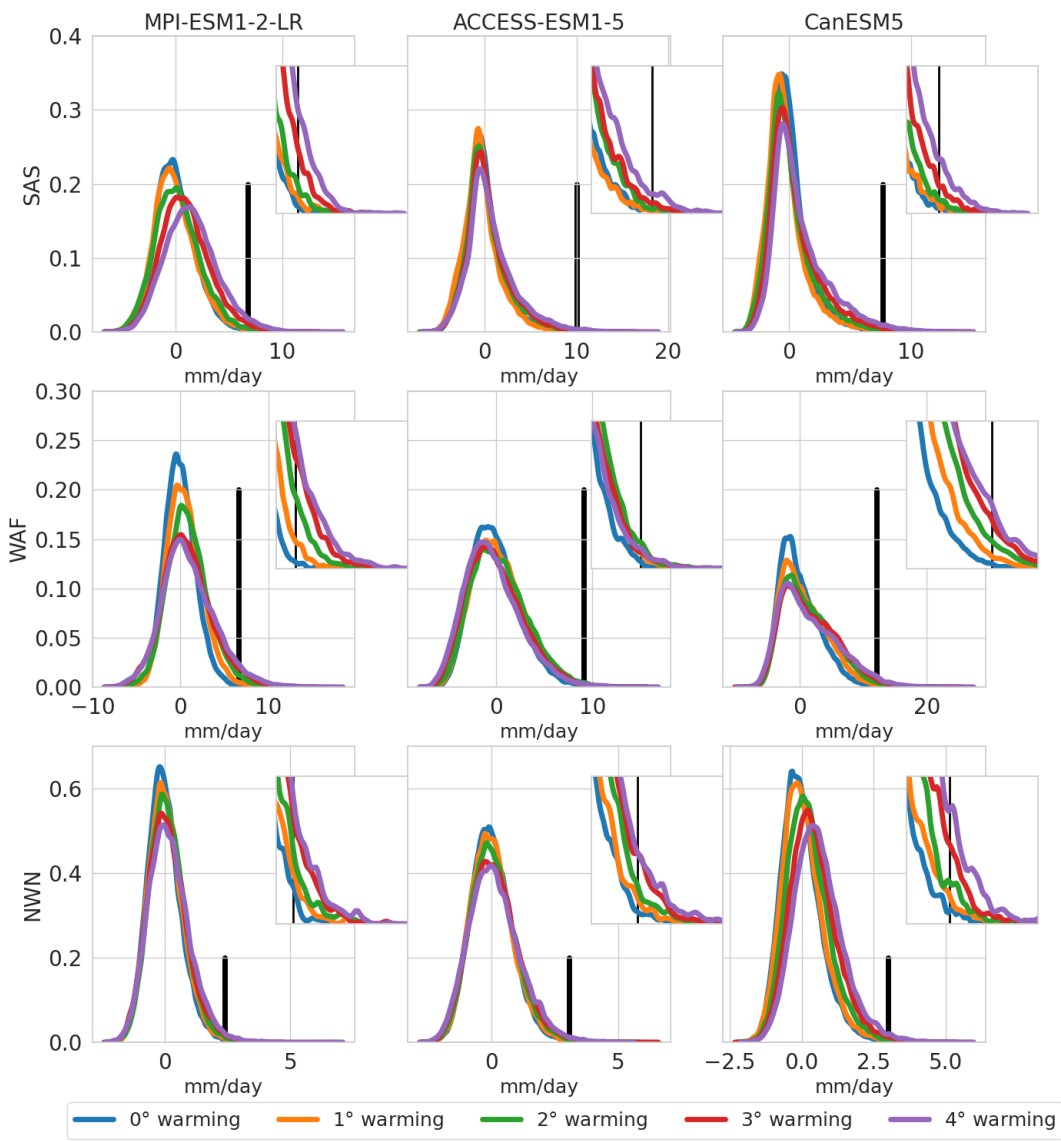

**Figure 8.** PDFs of total changes in JJA precipitation over South Asia (SAS), West Africa (WAF) and North West North America (NWN) under different global warming levels for all three models. Inserts show the upper tail of the distributions and the black horizontal line indicates the 0.999 quantile threshold.

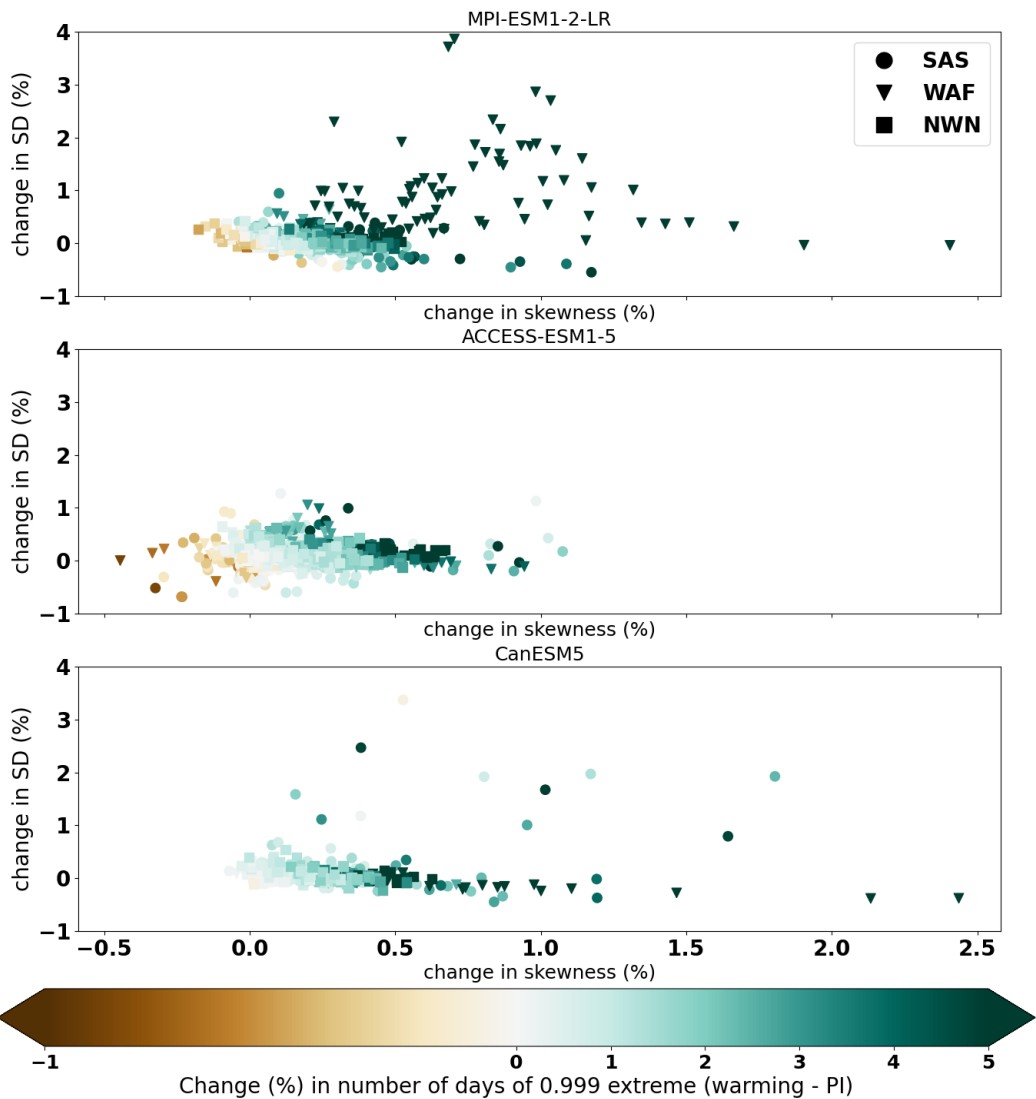

**Figure 9.** Regional changes in the SD and skewness for extreme JJA precipitation events are shown for SAS, WAF and NWN for each of the three models. Each marker in the figure represents one grid point.

the model uncertainty space, whereas SMILEs only sample uncertainty from climate internal variability. Their moisture budget analysis reveals that changes in variability are driven by changes in vertical moisture advection and thermodynamics. Similar conclusions are drawn by Zhang et al. (2024) using the observed increase in precipitation variability in ERA5. On longer time scales there might be link to ENSO variability suggested by Kohyama and Hartmann (2017).

Most regions where changes in the mean dominate the summertime precipitation variations are located in the SH. The influence of seasons plays a significant role in this hemispheric asymmetry. Figures B1 - B4 illustrate similar results for the NH during NH winter (DJF). Some of the observed changes are related to seasonal shifts. For example, there is an increase in intense precipitation due to changes in variability during DJF while the number of these days decreases in JAS in in South America (excluding the Amazon region) and South Africa. In Southeast Asia similarly, the number of intense precipitation events decreases due to changes in variability. However, only the MPI-ESM1-2-LR model shows that changes in the mean dominate future changes in wintertime intense precipitation over Southeast Asia.

Some extreme attribution studies follow the method from Philip et al. (2020) which assumes that the shape of the distribution stays constant. However, we find that this assumption cannot be made in a future climate, although the exact distribution changes remain uncertain due to large discrepancies between CMIP6 models. Nonetheless, our findings highlight the importance of including daily variability in climate change impact and attribution studies. While we find small or no change in the summertime mean precipitation, a clear increase in the number of extreme precipitation days is evident. Therefore, impact studies which only concentrate on the mean climate would inevitably underestimate the effects of extreme events.

This is further applicable to, for instance, the development of statistical emulators. Most emulators only consider global-mean temperature or precipitation effects or apply simple linear scaling (Nath et al., 2021; Watson-Parris et al., 2021). Based on the findings from this study, we recommend that the training of emulators should include training with daily weather variability to capture the complete climate change impacts. Furthermore, more work is needed such that emulators and simple climate models can fully simulate the effects of different climate drivers, as already highlighted by Persad et al. (2023). For future applications, it is relevant to know how well present ESMs can replicate observed daily climate variability. Lund et al. (2023) shows that the MPI-ESM1-2-LR and CanESM5 model capture the mean present-day precipitation rates well. However, evaluating the accuracy of climate models in predicting present-day extreme events is challenging due to sparse observational data. With only three rare extreme events recorded (based on our definition), the limited dataset hampers robust model validation, leading to uncertainty in the model's ability to reliably reproduce such rare but impactful occurrences. As different models show different kinds of underlying distributions, the limitation of this study is the small number of ESM ensembles used.

Although our findings primarily focus on the impact of climate change on wet extremes, it is essential to note that changes in both mean and variability can enhance or reduce the occurrence of dry extreme events as well. When examining total changes occurring due to changes in the variability and mean, we found that changes in the mean reduce the likelihood of dry extremes while changes in variability exacerbate changes in wet extremes. This finding underlines that it is crucial to recognize that although changes in climate variability can influence the frequency of extreme events, these effects may be offset by shifts in the mean climatic conditions for dry extremes.

## 5 Conclusions

This study investigates the role of changes in mean and variability separately on daily summertime precipitation and maximum temperature for three different large-ensemble CMIP6 models. We focus on changes under four different global warming levels ($1 - 4°C$) as well as changes in the near future driven by different anthropogenic drivers (specifically anthropogenic aerosols and greenhouse gases).

Our main findings are listed below:

– Changes in daily variability are the main drivers of changes in the likelihood of extreme summertime precipitation. In contrast, the change in the mean state is the primary driver of changes in temperature.

– Three key regions, namely Asia, Arctic and Sub-Saharan Africa, show that changes in the width and shape of the PDFs are particularly relevant in influencing summertime precipitation.

– In the near future, aerosol emission reductions are likely to increase the likelihood of extreme summertime precipitation over Asia.

– Model discrepancies dominate estimates of the impact of different climate drivers in the near future.

We find that aerosol emissions play a key role in the near-future evolution of regional precipitation extremes due to the ongoing reduction of anthropogenic aerosol emissions and their strong influence on daily precipitation variability. This would suggest that simple aerosol representations, as is implemented in the MPI-ESM1-2-LR model, lead to an underestimation of aerosol impacts compared to models with more advanced aerosol schemes, like in ACCESS-ESM1-5. Still, large uncertainty remains on how regional PDFs of precipitation will change (shape and width) in the future under different emission pathways. Global warming will lead to more extreme precipitation in many regions. How the near-term mix of anthropogenic and natural drivers will influence the width and shape of the distributions of daily weather, however, is still a relevant topic for future research.

*Code availability.* All codes used in this study can be accessed via https://github.com/kallenordling/variability

*Data availability.* Data used in this paper is available from Nordling (2024)

# Appendix A: Results from PDRMIP

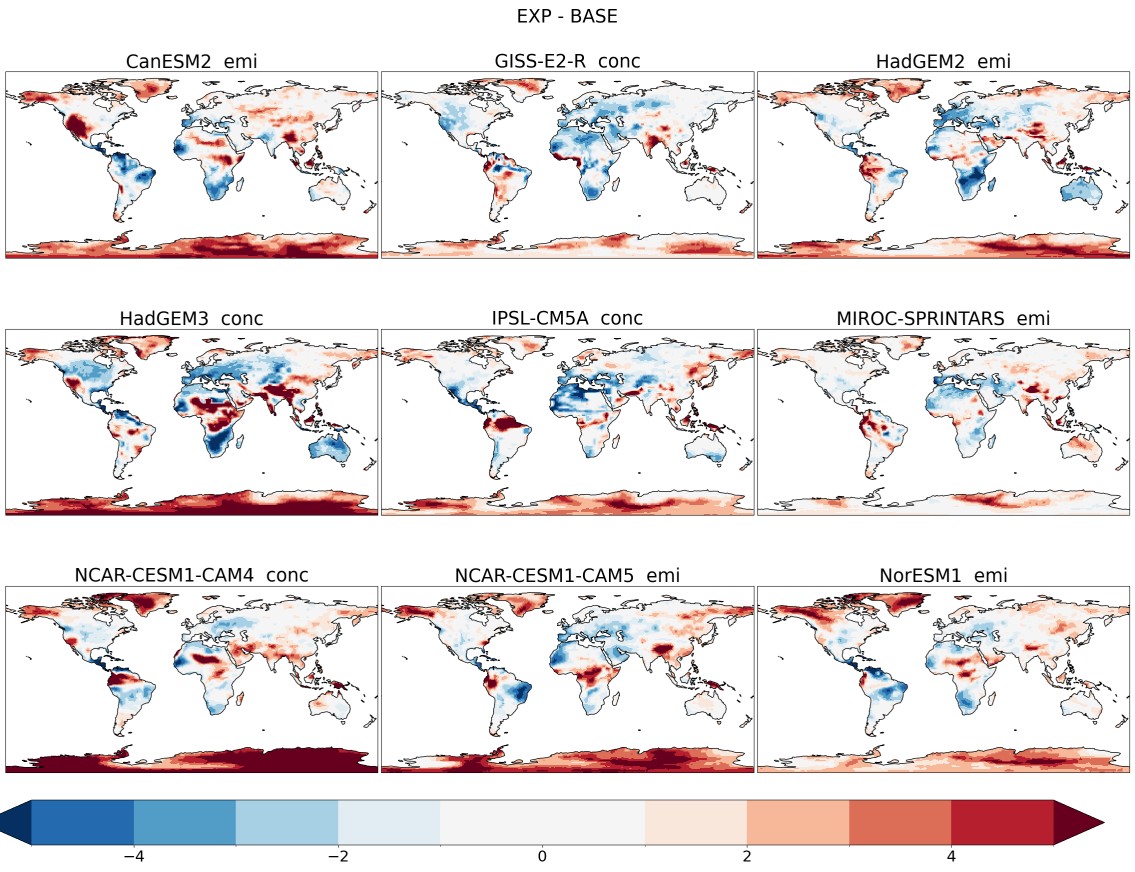

**Figure A1.** Changes in the average number of days per year of extreme (0.90 quantile) precipitation due to the global doubling of $CO_2$ concentrations as simulated by nine different PDRMIP models. Panel titles indicate if a model is emission- (emi) or concentration-driven (conc).

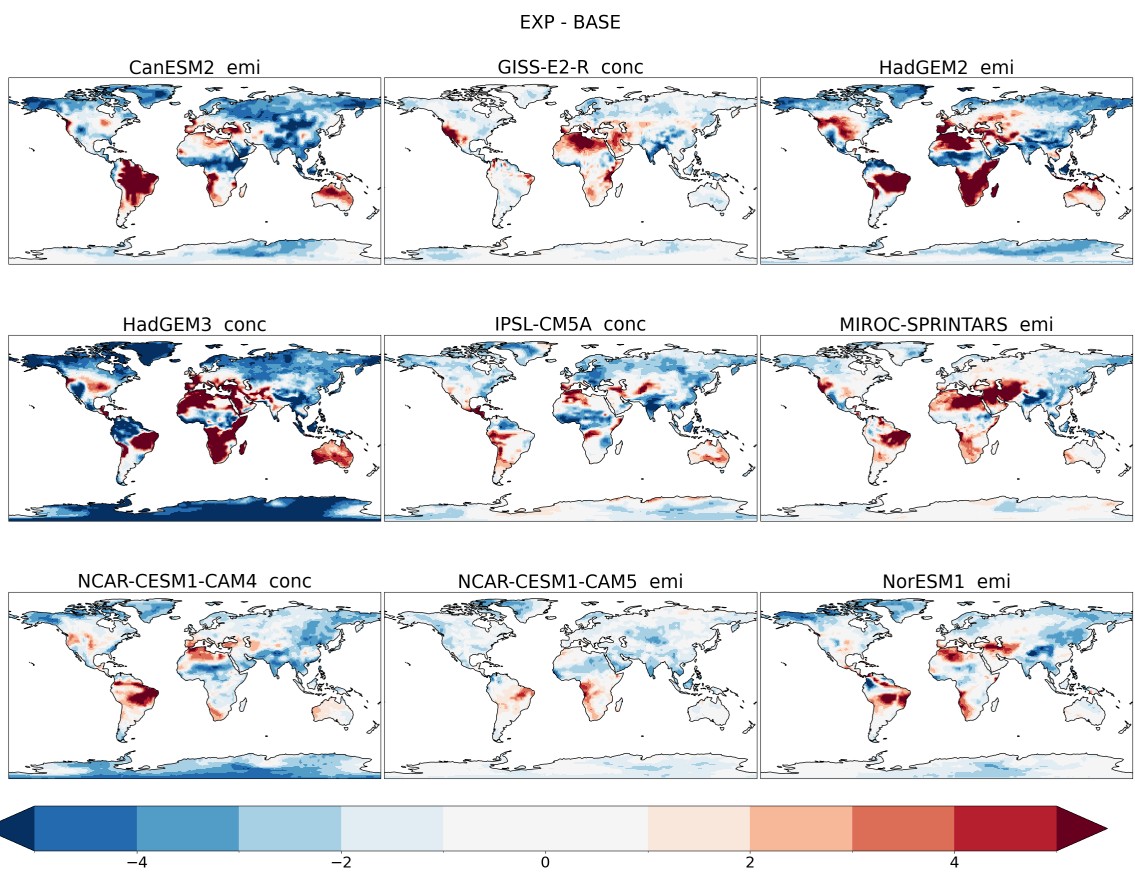

**Figure A2.** Changes in the average number of days per year of extreme (0.90 quantile) precipitation due to a global five-fold increase in sulfate emissions as simulated by nine different PDRMIP models. Panel titles indicate if a model is emission- (emi) or concentration-driven (conc).

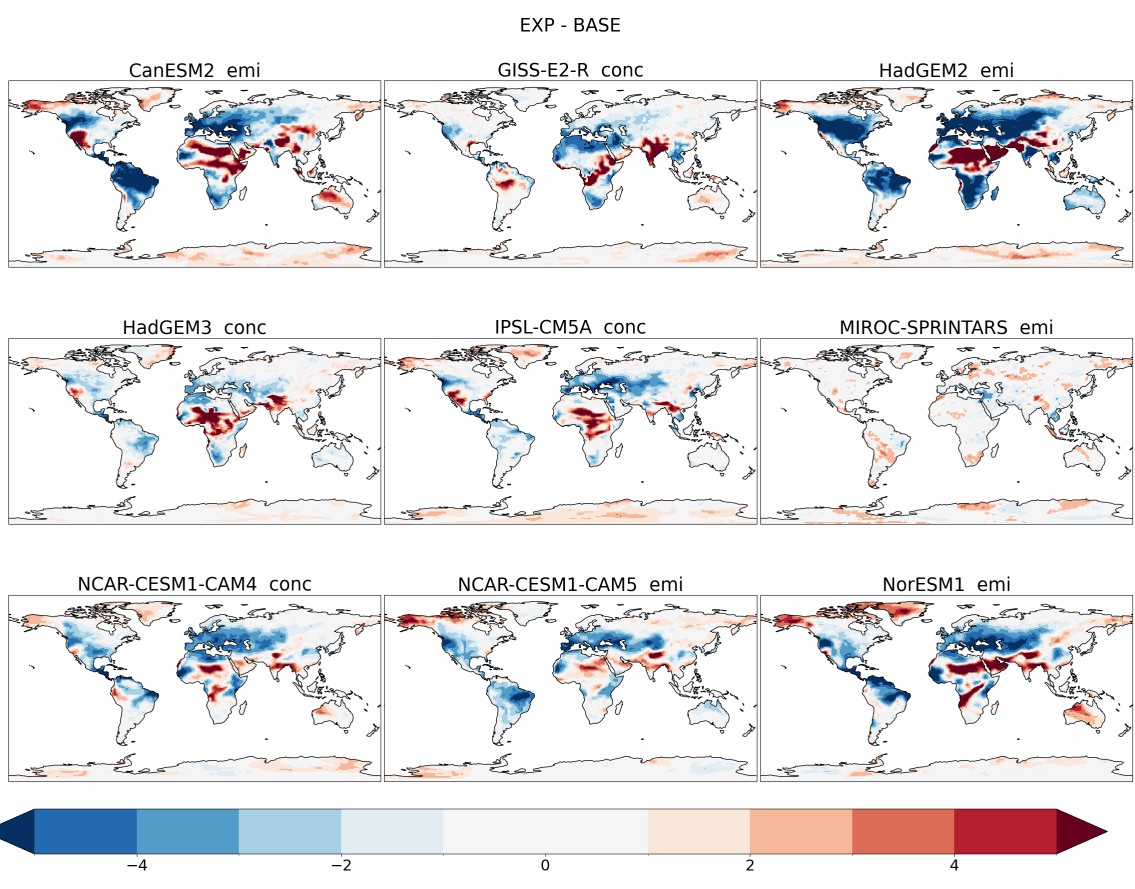

**Figure A3.** Changes in the average number of days per year of extreme (0.90 quantile) precipitation due to a global ten-fold increase in black carbon emissions as simulated by nine different PDRMIP models. panel titles indicate if a model is emission- (emi) or concentration-driven (conc).

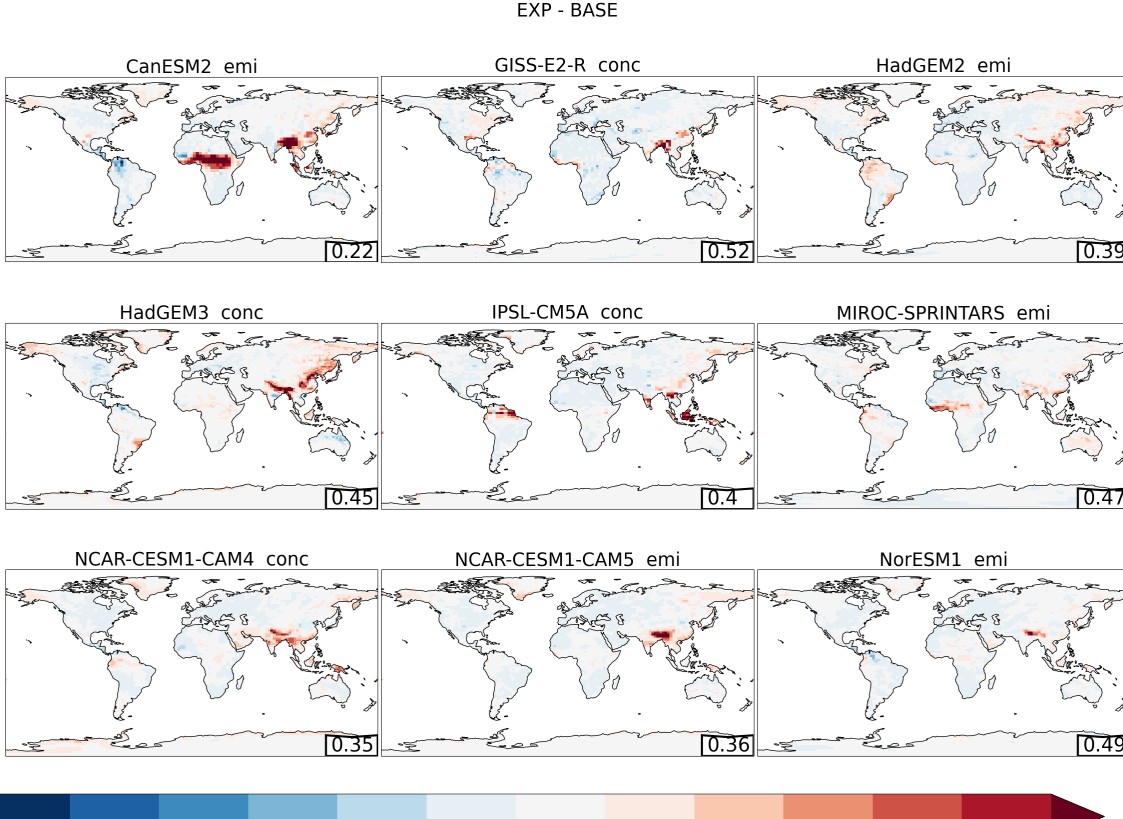

**Figure A4.** Changes in the daily summertime PDF standard deviation due to a global doubling of $CO_2$ concentrations as simulated by nine different PDRMIP models. panel titles indicate if a model is emission- (emi) or concentration-driven (conc). Correlation between standard deviation and change in extremes is shown in the corner.

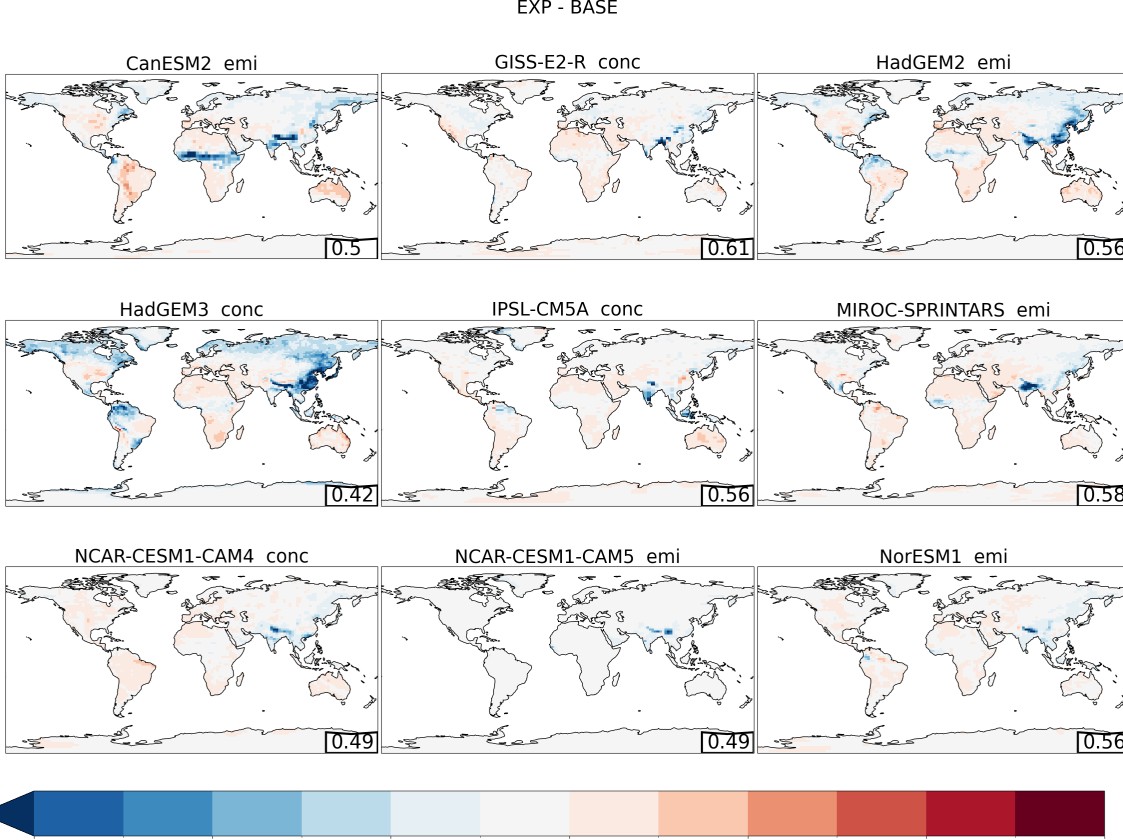

**Figure A5.** Changes in the the daily summertime PDF standard deviation due to a global five-fold increase in sulfate emissions as simulated by nine different PDRMIP models. panel titles indicate if a model is emission- (emi) or concentration-driven (conc). Correlation between standard deviation and change in extremes is shown in the corner.

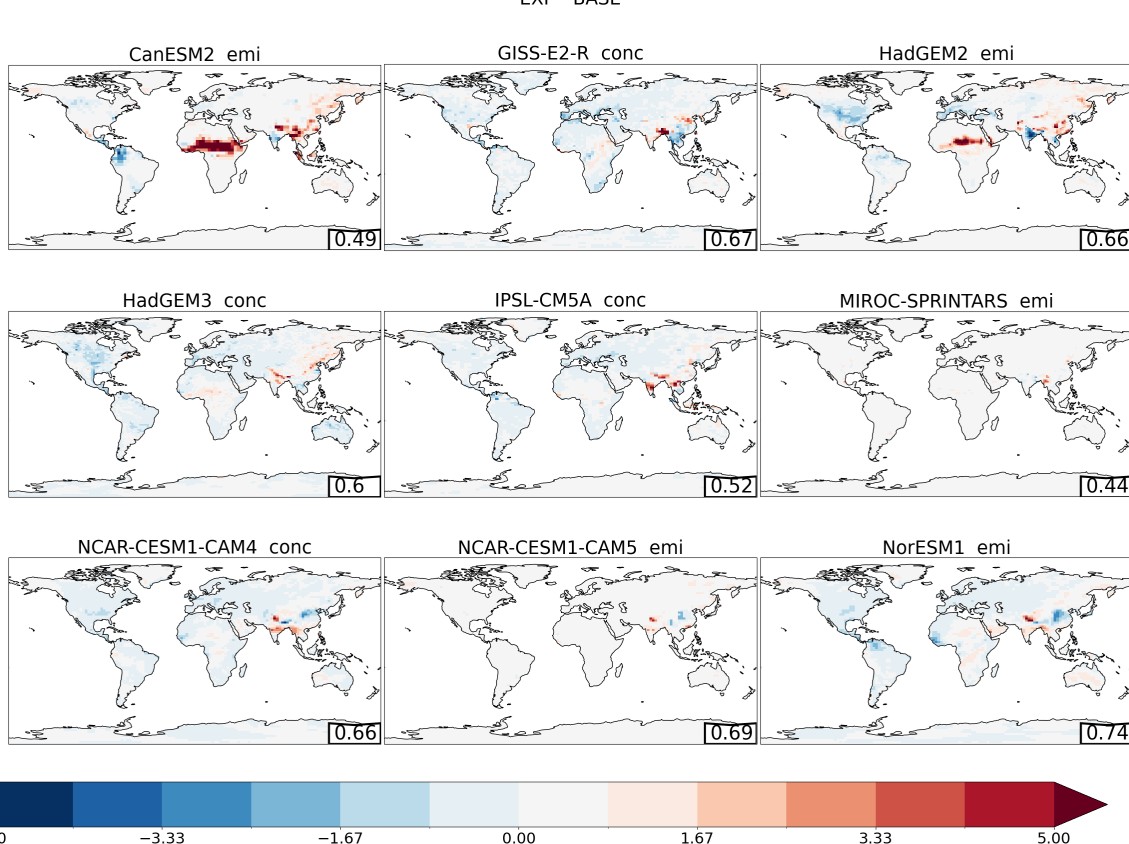

**Figure A6.** Changes in the daily summertime PDF standard deviation due to a global ten-fold increase in black carbon emissions as simulated by nine different PDRMIP models. panel titles indicate if a model is emission- (emi) or concentration-driven (conc). Correlation between standard deviation and change in extremes is shown in the corner.

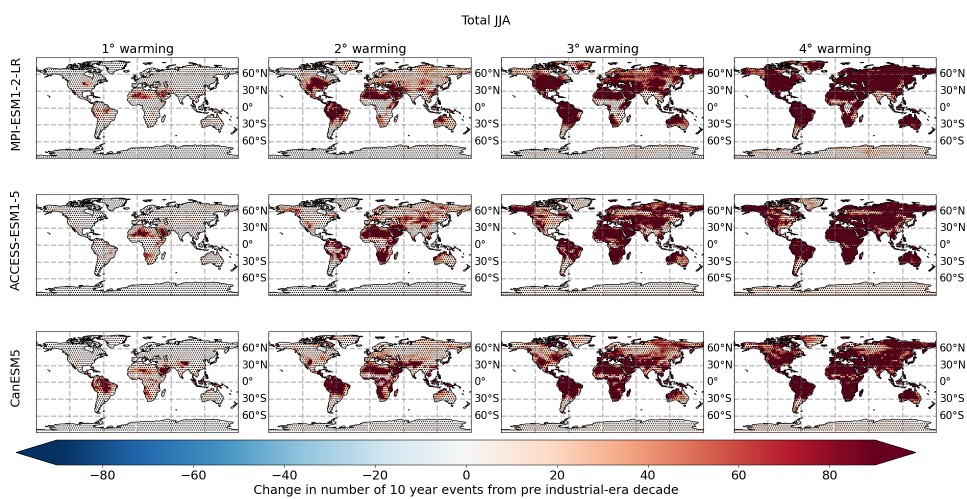

**Figure A7.** Change in number of extreme heat days due to change in global warming levels.

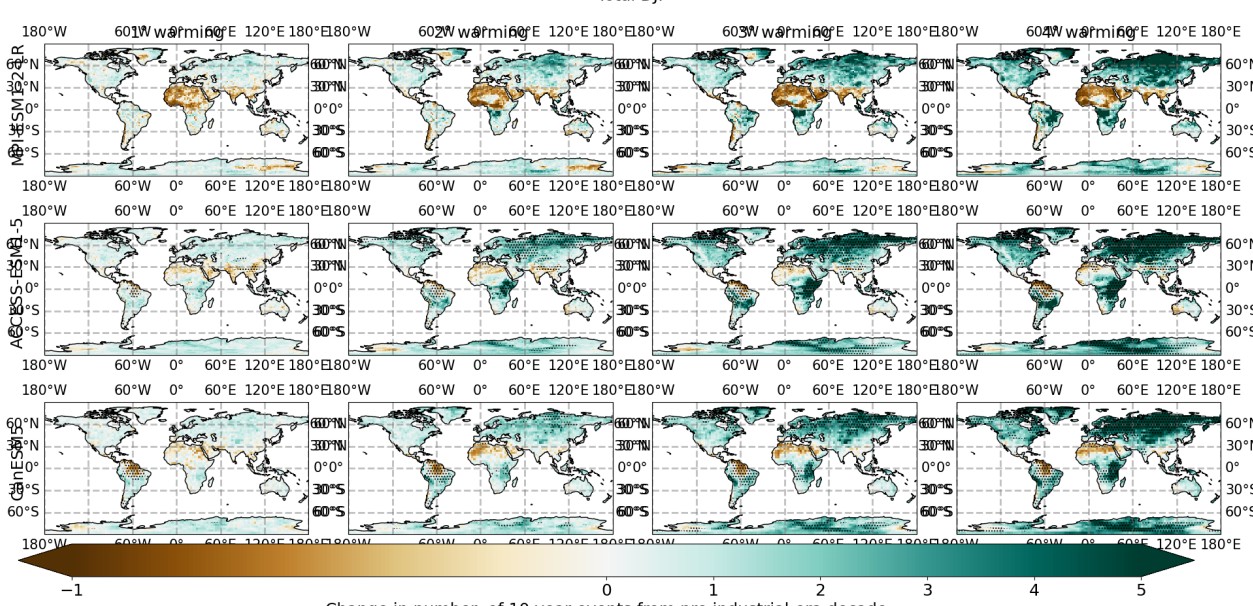

**Figure B1.** Total change in the number of days of intense precipitation events during DJF under different global warming levels. Stippling indicates regions where changes in PDFs are significant at p > 0.05

**Appendix B:  Main figures for DJF**

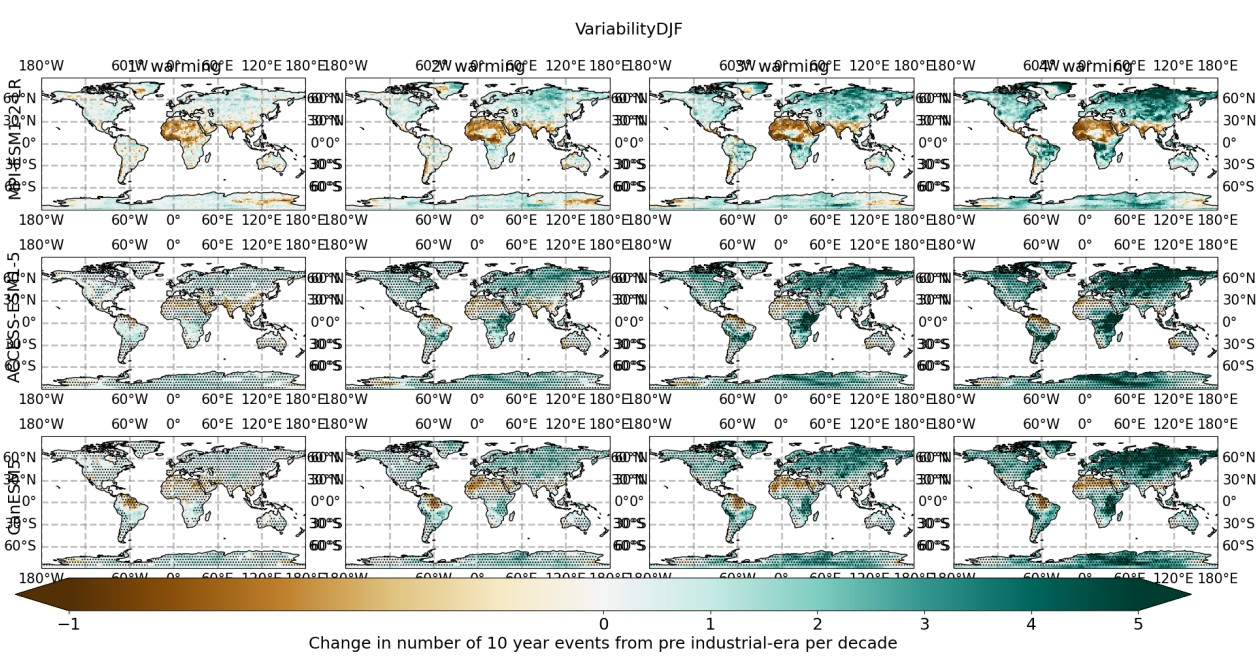

**Figure B2.** Changes in the number of days of intense DJF precipitation events due to changes in variability under different global warming levels. Stippling indicates regions where changes in PDFs are significant at p > 0.05

change in mean DJF

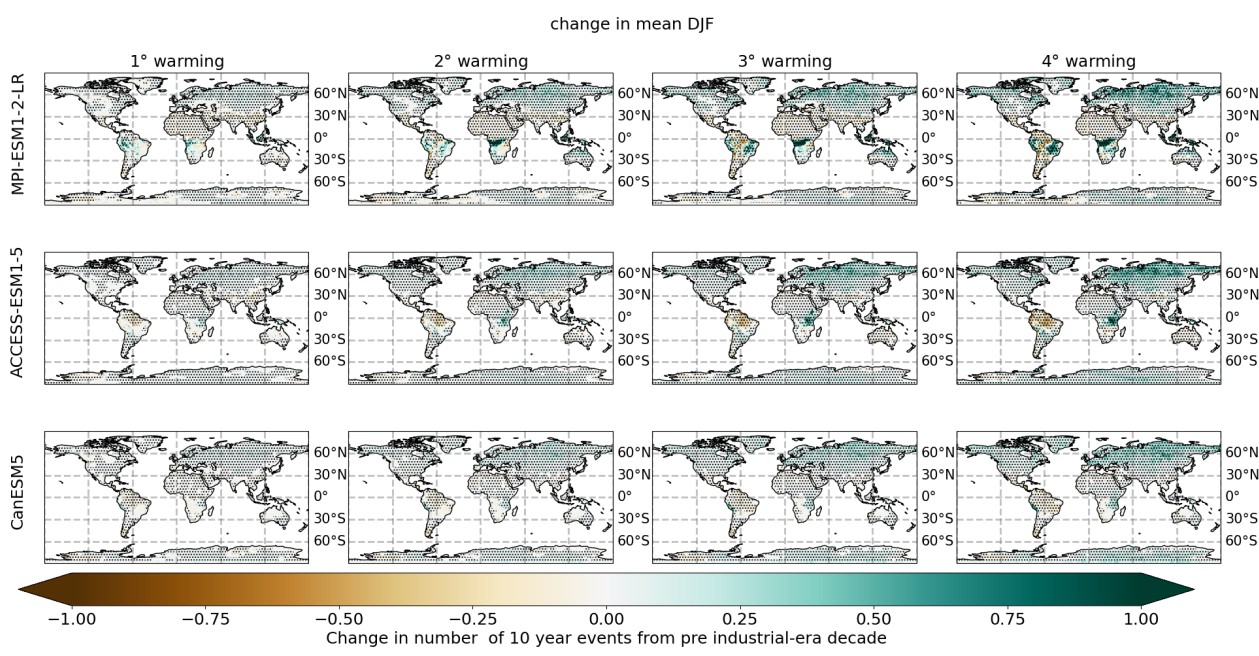

**Figure B3.** Changes in the number of extreme DJF precipitation events due to changes in the mean under different global warming levels. Stippling indicates regions where changes in PDFs are significant at p > 0.05

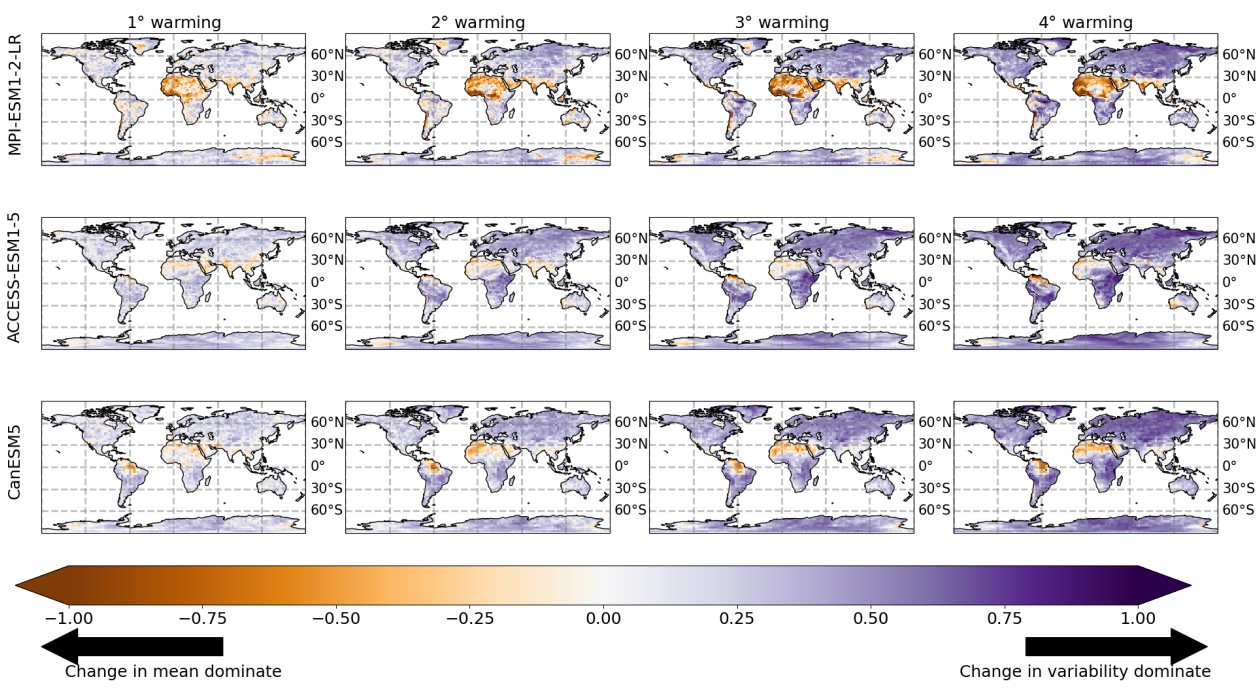

**Figure B4.** Decomposition of regional changes in DJFprecipitation and daily maximum temperature extremes into changes in the mean and changes in variability under four warming levels (columns). Figure shows mean of three models and hatching indicates regions where all three models do not agree. Orange colors indicate regions where the change in the mean dominate changes in the extremes and purple colors indicate regions where variability dominates.

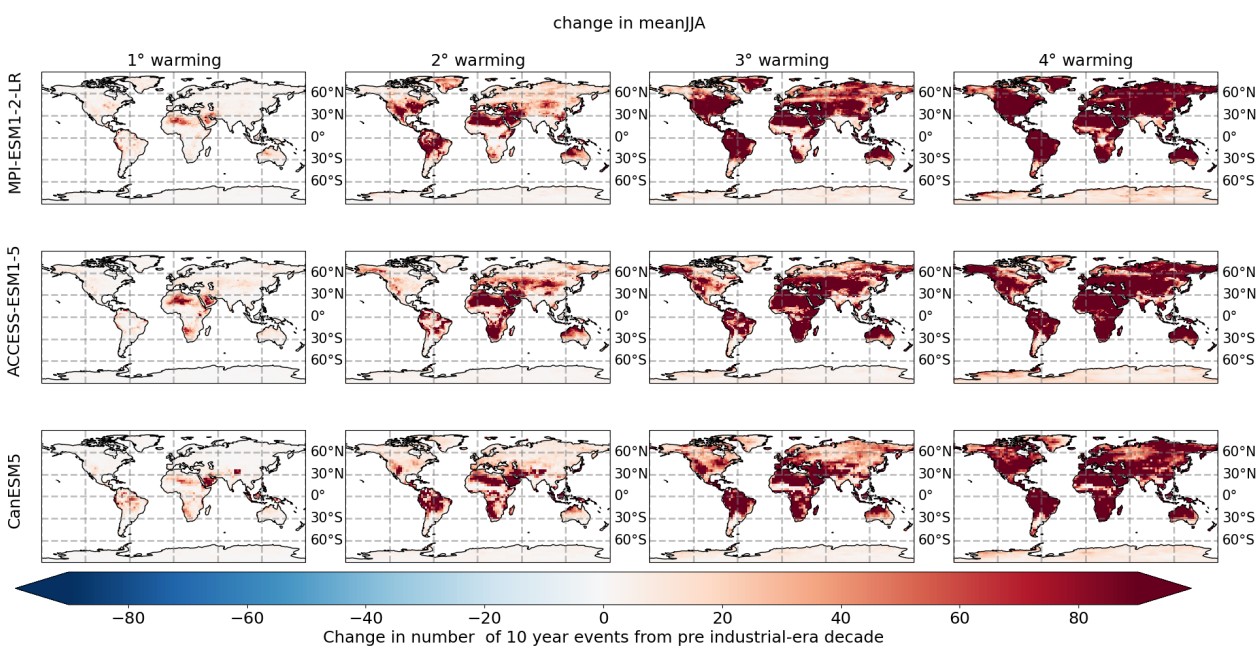

**Figure B5.**

## Appendix C: Near future

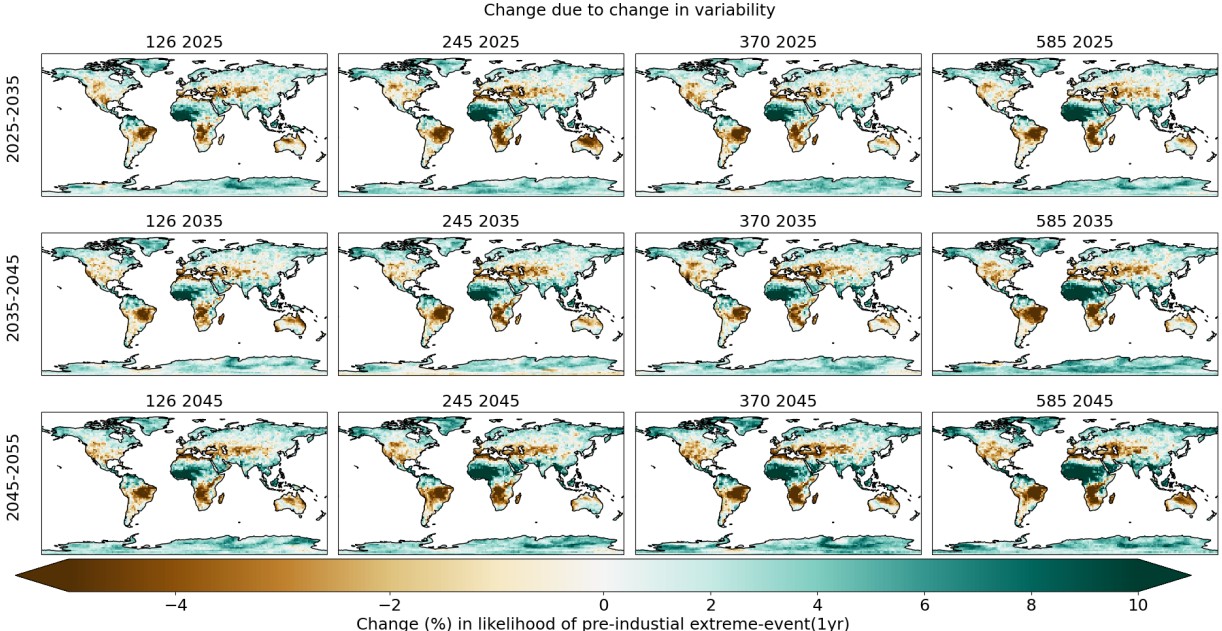

**Figure C1.** Near-future changes in number of extreme precipitation days for MPI-ESM1-2-LR under four different SSP scenarios (columns) and for three different time periods (from left to right: 2025–2035, 2035–2045 and 2045–2050).

## Appendix D: Model discrepancies

*Author contributions.*  All authors contributed to the writing. BHS came up with the original concept, NLSF came up with the concept of extremes and KN performed the analysis.

*Competing interests.*  The authors declare no competing interests.

*Disclaimer.*  TEXT

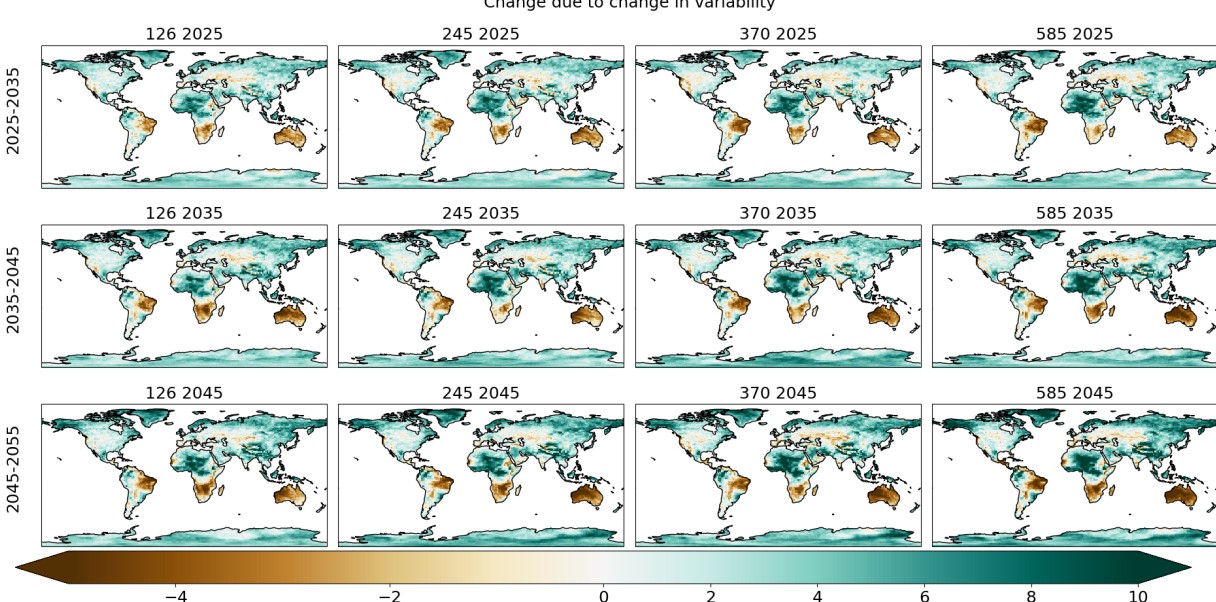

**Figure C2.** Near-future changes in number of extreme precipitation days for ACCESS-ESM1-5 under four different SSP scenarios (columns) and for three different time periods (from left to right: 2025–2035, 2035–2045 and 2045–2050).

*Acknowledgements.* We acknowledge support by the Research Council of Norway [Grant no. 324182 (CATHY)], and the Center for Advanced Study in Oslo, Norway that funded and hosted HETCLIF centre during the academic year of 2023/24. We also acknowledge the resources provided by UNINETT Sigma2 – the National Infrastructure for High Performance Computing and Data Storage in Norway (project account NS9042KK). We also acknowledge the Academy of Finland grant no. 337552.

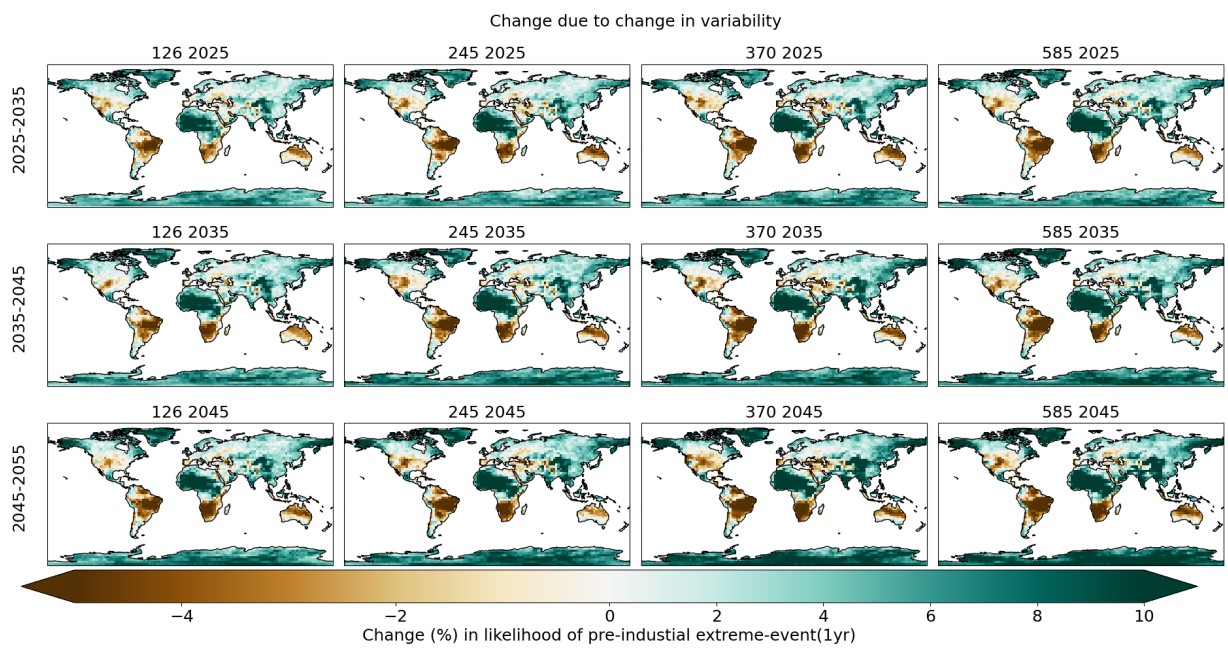

**Figure C3.** Near-future changes in the number of extreme precipitation days for CanESM5 under four different SSP scenarios (columns) and for three different time periods (from left to right: 2025–2035, 2035–2045 and 2045–2050).

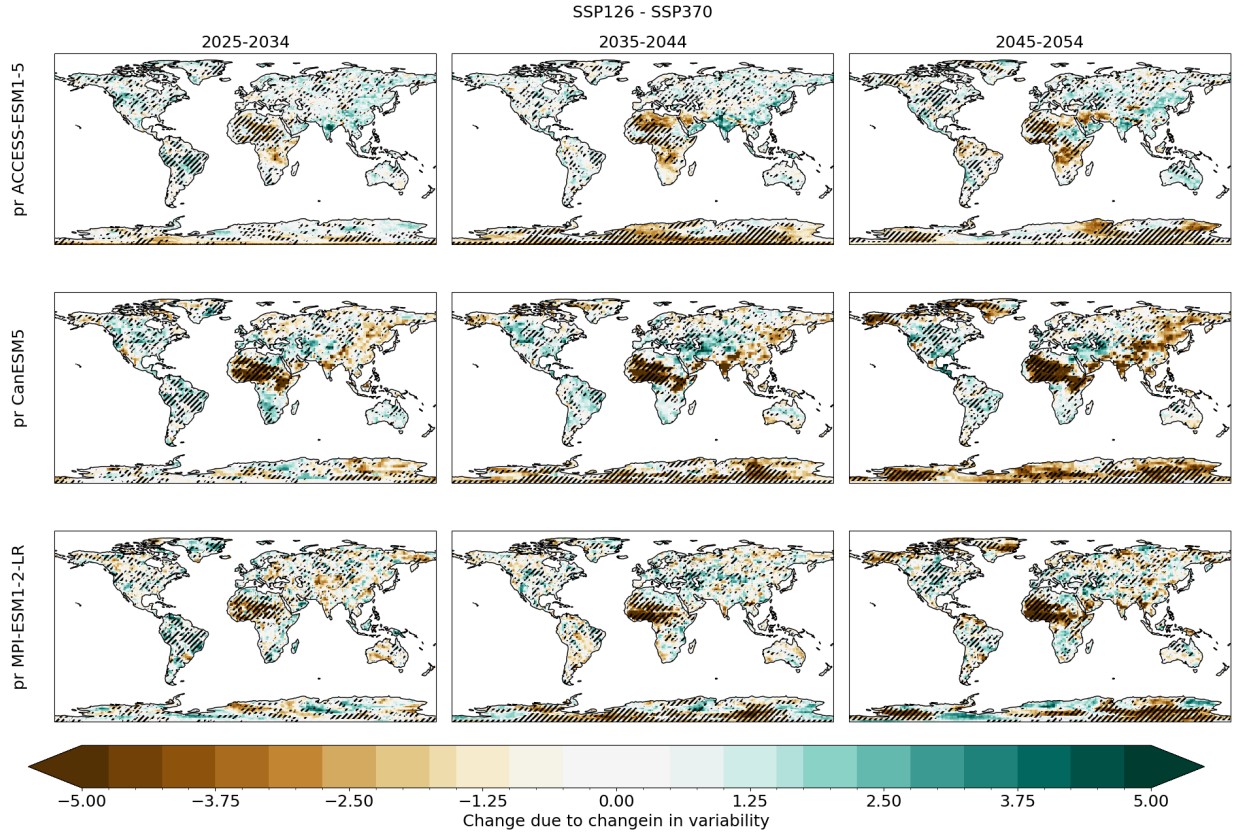

**Figure C4.** Change in likelihood in days of extreme JJA precipitation between SSP1-2.6 and SSP3-7.0 for three different models ACCESS-ESM1-5 (Row 1), CanESM5 (row 2) and MPI-ESM1-2-LR (row 2). Hatching indicates regions where all three models agree on the sign of the change.

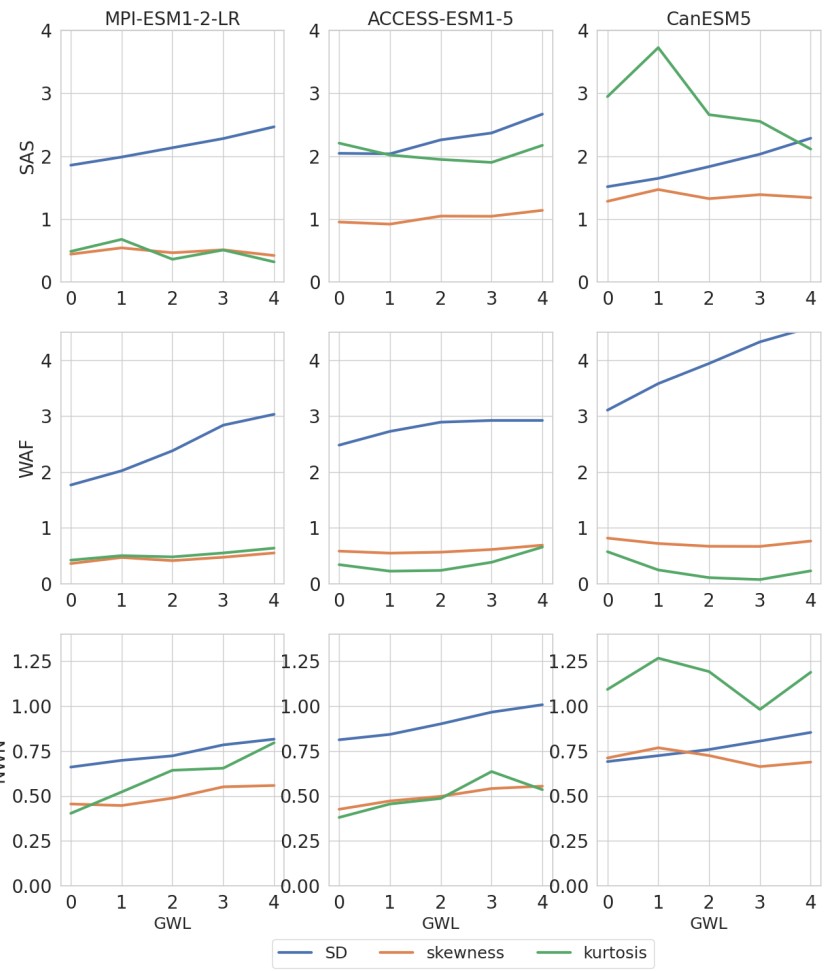

**Figure D1.** Evolution of regional mean standard deviation, kurtosis and skewnes for three regions, and three models.

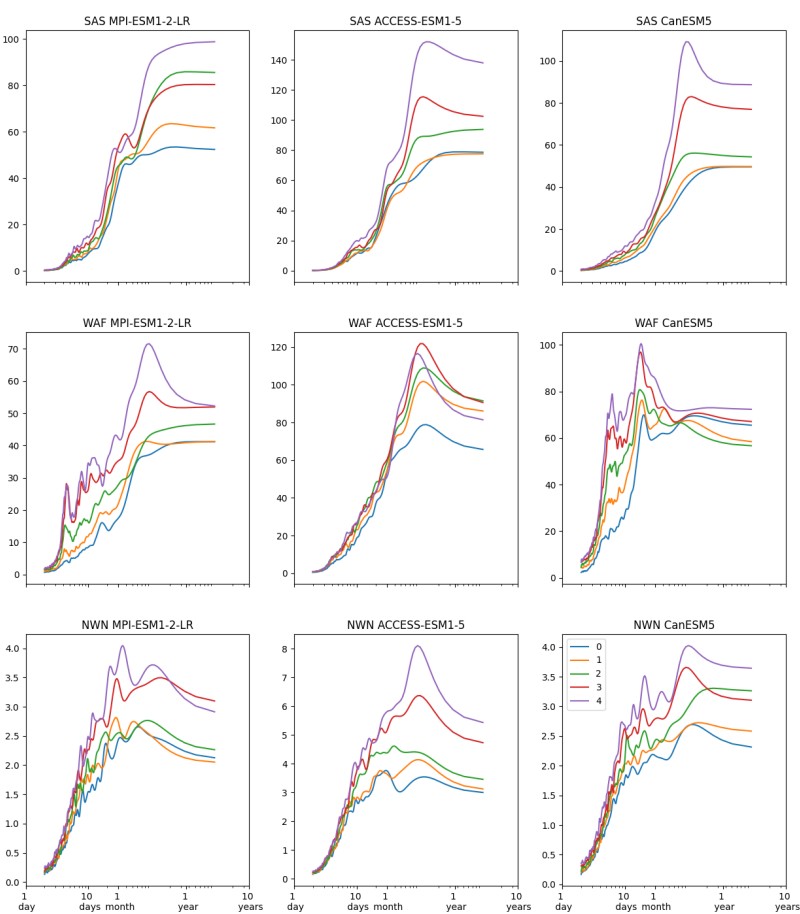

**Figure D2.** Evolution of regional power spectral density for three regions, and three models.

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
