# Peer review of "Climate variability can outweigh the influence of climate mean changes for extreme precipitation under global warming"

_EGUsphere, 2024_

## Referee Comment (RC1)

Review of "Climate variability can outweigh the influence of climate mean changes for extreme precipitation under global warming"

**General comments:**

This paper investigates the change in number of extreme precipitation and daily maximum temperature days at various global warming levels and time periods in response to various forcings. The authors show the increase in extreme precipitation days is generally associated with changes in the shape of the distribution whereas daily maximum temperature is generally associated with a shift in the distribution. The details of the spatial pattern of these responses depends on models, possibly due to differences in representation of aerosols and climate sensitivity.

The analysis and findings presented are interesting and a useful addition to the literature on the response of precipitation and temperature extremes to various forcings. A comprehensive analysis like this is useful and I think the manuscript can be further improved if the authors better motivate the reasons for looking at different global warming levels, SSPs at different decades, and different forcings. Specifically, while the current introduction describes the literature in detail, it lacks a clear statement on what is unknown and a research question the manuscript addresses. I suggest the authors add a paragraph in the introduction dedicated to this. I also have several specific comments below that I would like to see addressed prior to publication.

**Specific comments:**

Line 65: What is the reason for choosing JJA instead of looking at JJA in the NH and DJF in the SH? The subsequent figures show that several responses are hemispheric asymmetric (e.g., wetting in Northern Africa vs drying in Southern Africa, role of mean change in extreme precip appears mostly in the Southern Hemisphere). Is this asymmetry due to the seasonal asymmetry in NH and SH? I think a discussion on this and showing the DJF equivalents of the main figures in the appendix would be important.

Line 81-83: I think it would be better to introduce the definition, "PDF of the total change", after the discussion of removing the annual cycle. I expected the leftmost panel of Fig. 1c to be a PDF of temperature before removing the annual cycle because the sentence describing the removal came after the definition of the "PDF of the total change".

Line 81-86: This section is confusing to read because of the lack of detail of the kind of mean or variability the authors have in mind. For example, in the sentence "The second step involves removing the annual cycle at each grid point for each GWL which gives a PDF that only differs in variability", I believe the authors are saying that by removing the annual cycle, the resulting PDF quantifies *daily* variability. It would be helpful to the reader if the frequency or spatial details of the mean and variability are specified here and elsewhere in the paper. If the same kind of mean or variability is used hereafter, the authors can define this once here.

Line 86: How are the changes in standard deviation and skewness isolated and quantified?

Fig. 1: Considering that the paper focuses mostly on precipitation, would it not be preferred to show this example using precipitation?

Fig. 1b: Is this for an example grid point or the global mean? What is the location? What do the different colored lines represent? Do they represent hot, median, and cold percentile temperature? Can you add a legend or describe it in the caption?

Line 126: Is this because there is only one ensemble member per model for PDRMIP simulations? If so, it would be helpful to state this explicitly here.

Fig. 2: Is there a reason for highlighting these 3 models in particular? NorESM1 appears to be highlighted because the response to CO2 is the smallest but the reason for highlighting the other two models is not provided.

Line 144: What is the significance level and is the statement here that the spatial patterns of changes in SD and number of extreme days significantly different or correlated?

Line 146-147: I'm confused by this interpretation. How does a higher correlation between changes in SD and extremes evaluated separately for different forcings tell us how the PDF change is more dependent on changes in aerosol vs CO2? Since SD is influenced by changes in both low and high precip extremes, my interpretation of this result is that the response to

aerosol forcing disproportionately affects high vs low precip extremes and thus is better captured by SD, whereas the response to CO2 forcing may also be affecting low precip extremes.

Fig. 3 caption: Do you mean $p < 0.05$?

Line 165: It looks like an increase in the number of intense precipitation events due to daily variability change is not seen everywhere in Asia. For example in MPI-ESM1-2-LR, there is a decrease along the 30th parallel north and for ACCESS-ESM1-5 and CanESM5, there is a weak response in the Middle East and Central Asia. I suggest you specify the subregions within Asia where the response is large.

Fig. 4 and 5: Is there stippling anywhere in these plots? I don't see them even when I zoom into individual subpanels.

Fig. 5: It would be easier to compare the role of mean vs variability change if the same colorbar settings were used for both Fig. 4 and 5. Currently the colorbar in Fig. 4 saturates on the positive end at 5 whereas the colorbar in Fig. 5 saturates at 1.

Line 175 and Fig. 6b: I'm surprised the change in mean dominates in Eastern Brazil, Southern Africa, and Northern Australia. Looking at Fig. 4 and 5 it appears to me that the brown is a darker shade in Fig. 4 compared to Fig. 5 for both MPI and CanESM. Can you provide plots with the variability and mean components side-by-side?

Line 181: What does "rarely observed" mean? Is it the 0.999 quantile?

Line 188-191: So the results shown here are not following the methods described in line 87-88? If so, you should mention in the methods section that you consider two different quantiles of extremes for CMIP6 data.

Line 201-202: Since not all of the domain shows a decrease in days of extreme precipitation I suggest you specify the region (e.g., South, Southeast, and East Asia)

Line 203-204: What you are describing here is only apparent around the Tibetan Plateau. I suggest specifying this.

Line 205: What does significant mean here? Statistically significant? Significant for impacts?

Line 222-234: Do you follow a systematic process for labeling a distribution as Gaussian, Gamma-like, or exponential-like? Otherwise characterizing the distributions like this is subjective and arbitrary. What are the standard deviation, skewness, and kurtosis of these distributions?

Line 231: Have you quantitatively tested the similarity between different distributions? (e.g., Kolmogorov Smirnov test)

Line 235-239: Can you discuss the regional dependence of the relative role in SD vs skewness change? For example in MPI, SD change is particularly large in WAF. In CanESM, SD change is large in SAS.

**Technical comments:**
Line 27: "unceratinties" → "uncertainties"

Line 121: "asses" → "assess"

Line 144: "test test" → "test"

Line 160 and 175: "South Africa" → "Southern Africa"; South Africa is a country whereas Southern Africa refers to the southern half of the African continent. I assume you mean the latter vs the former.

Line 171: "northern America" → "North America"

Fig. 9 colorbar label: "extream" → "extreme"

Line 257: "underplaying" → do you mean "underlying"?

Line 282: remove "net"

---

## Referee Comment (RC2)

**Title:** Climate variability can outweigh the influence of climate mean changes for extreme precipitation under global warming

**General comments:**

The authors have highlighted that changes in precipitation variability may contribute more significantly to the frequency of extreme precipitation events than changes in mean state of precipitation under global warming scenarios in fully coupled Earth System Models (ESMs) of CMIP6. Additionally, they suggest that the reduction of aerosol emission could potentially increase the frequency of extreme summertime precipitation events over Asia, though there remains some model uncertainty.

This manuscript is well-organized, and provides valuable insights into the connection between changes in precipitation variability and the frequency of extreme precipitation events under global warming. It also highlights the importance of aerosols in the frequency of Asian extreme precipitation events under global warming scenarios. However, the manuscript has some unclear points that could benefit from further clarification. I hope my comments will help enhance the quality of this manuscript.

**Specific comments:**

1. L33–34:

Could you clarify how Asian aerosol emissions affect Arctic temperatures and the Australian monsoon? Local aerosol emission could result in direct radiative forcing through advection and could occur teleconnection pattern through forced Rossby waves. It would be great to clarify this part to justify next sentence about the impact of aerosol emission on local and remote regions.

2. L75–90:

This part is a little bit confusing, making It difficult to understand the method. I have a few questions and suggestions for clarification:

1) What is meant by "single-model initial-condition large ensembles" in this section?

2) From "the second step ~ ", did you remove the annual cycle from your definition of 'variability'?

3) You defined an extreme event as one that exceeds the $0.999^{th}$ quantile (~$99.9^{th}$ percentile) in one instance and then used a different definition of extreme events as those that exceed $99^{th}$ percentile elsewhere.

I recommend clarifying this part of the methodology for the readers.

3. L130–138:
The authors described the overall features of daily precipitation variability driven by anthropogenic forcings. However, they presented results from only three models and particularly highlight NorESM, which has the weakest response of variability to doubled CO2 levels. Is there specific reason why the authors chose to display results from only these models?

Furthermore, the authors mentioned that "these changes correlate with changes in the SD.", providing only a range of correlation coefficients from 0.22 to 0.49. I am curious about which region's variability could be explained by changes in the SD. I recommend including a spatial map to show the correlation between changes in SD and variability.

4. L157–167:
The authors highlighted the important role of precipitation variability in the frequency of extreme precipitation events. Do you have any thoughts on which timescale of variability (or which physical phenomena) is associated with this?

5. L182–184:
Is there figure for this example about ACCESS-ESM1-5?

6. L196–199:
This part is difficult to understand, specifically how the authors extracted the effect of aerosols from the difference between SSP3-7.0 and SSP1-2.6. Could you explain more details in why this difference represents the effect of aerosols?

7. L209–216:
Does this part mean that CanESM5 and ACCESS-ESM1-5 show more reasonable response of precipitation extreme frequency to aerosol emission change?

8. L220:

Could you explain why the authors chose these regions?

9. L226 –230:
ACCESS-ESM1-5 doesn't seem to be widen across different GWL.

**Technical corrections:**
1. L14:
2020-2040 → 2020–2040

2. L28, L29:
Chen et al. (2021) → (Chen et al., 2021)
Samset (2022) → (Samset, 2022)

3. L31, L33:
emissions(… → emissions (
temperatures(… → temperatures (

4. L82:
' ' → "

5. L97:
timeperiod → time period

6. L121:
asses → assess

7. L124:
Typo? year2000

8. L144:
test test → test

9. L191:
B2, B3 → B2, and B3

10. L237:
CanEMS5 → CanESM5

---

## Author Comment (AC1)

We thank the reviewers for their helpful comments for improving our manuscript. The referee comments are shown with *blue font color and italics*, and our point-to-point responses with standard font.

**1 Referee 1. comments**

**1.1 General comments**

I think the manuscript can be further improved if the authors better motivate the reasons for looking at different global warming levels, SSPs at different decades, and different forcings. Specifically, while the current introduction describes the literature in detail, it lacks a clear statement on what is unknown and a research question the manuscript addresses. I suggest the authors add a paragraph in the introduction dedicated to this

We have included the following paragraph to the introduction (L67): "What remains unclear is the role of variability: Are precipitation and temperature extremes becoming more severe due to changes in the mean state, or due to changes in day-to-day variability? Another uncertainty relates to the climate models themselves. Despite generally agreeing on the direction of changes in extreme precipitation, the current state-of-the-art climate models show significant uncertainty regarding the magnitude of these changes, especially at regional scales. In particular, the different implementations for anthropogenic aerosols and different climate sensitivities of different ESMs add to this uncertainty. Another gap in the current knowledge is how to translate the changes in the daily distribution of weather variability to meaningful quantities, like the number of extreme weather events."

**1.2 Specific comments**

Line 65: What is the reason for choosing JJA instead of looking at JJA in the NH and DJF in the SH? The subsequent figures show that several responses are hemispheric asymmetric (e.g., wetting in Northern Africa vs drying in Southern Africa, role of mean change in extreme precip appears mostly in the Southern Hemisphere). Is this asymmetry due to the seasonal asymmetry in NH and SH? I think a discussion on this and showing the DJF equivalents of the main figures in the appendix would be important.

We have appended the main figures for DJF to the appendix (Fig. B1-B4). Line 65 is changed to: "In this study, we focus on examine how daily variability in the Northern Hemisphere (NH) summer precipitation and daily maximum temperature is evolving under global warming and different emission scenarios. We also show the changes during NH winter months in the appendix Figure B1-B4."

The following section is added to the discussion: "Most regions where changes in the mean dominate the variations in summertime precipitation are located in the Southern Hemisphere. The influence of seasons plays a significant role in this hemispheric asymmetry. Figures B1-B5 illustrate similar results for the NH during winter (DJF). Some of the observed changes are related to seasonal shifts. For example, in South America (excluding the Amazon region) and South Africa, there is an increase in intense precipitation due to changes in variability, whereas the number of these days decreases during JJA. Similarly, there is a decrease in the number of intense precipitation events due to changes in variability in Southeast Asia. However, only the MPI-ESM1-2-LR model shows that changes in the mean dominate future changes in wintertime intense precipitation over Southeast Asia."

Line 81-83: I think it would be better to introduce the definition, "PDF of the total change", after the discussion of removing the annual cycle. I expected the leftmost panel of Fig. 1c to be a PDF of temperature before removing the annual cycle because the sentence describing the removal came after the definition of the "PDF of the total change".

Thanks for your suggestion. However, we think that introducing the term "PDF of total change" after removing the annual cycle could easily be confused with "PDF change due to changes in variability" which is why we decided to stick to the wording used previously.

Line 81-86: This section is confusing to read because of the lack of detail of the kind of mean or variability the authors have in mind. For example, in the sentence "The second step involves removing the annual cycle at each grid point for each GWL which gives a PDF that only differs in variability", I believe the authors are saying that by removing the annual cycle, the resulting PDF quantifies daily variability. It would be helpful to the reader if the frequency or spatial details of the mean and variability are specified here and elsewhere in the paper. If the same kind of mean or variability is used hereafter, the authors can define this once here.

For this section we have now defined that means and variability are daily means and daily variability

**Line 86: How are the changes in standard deviation and skewness isolated and quantified?**

We do not further isolate the changes in SD and skewness. Figure 9 only shows how the SD and skewness change relative to each other. We further mention in L93 that the changes in variability do include both, the changes in width (SD) and skewness.

**Fig. 1: Considering that the paper focuses mostly on precipitation, would it not be preferred to show this example using precipitation?**

Using measured precipitation data introduces additional noise to the figure and further we did not find a way to separate the effects of a change in skewness and standard deviation using measured data.

Fig. 1b: Is this for an example grid point or the global mean? What is the location? What do the different colored lines represent? Do they represent hot, median, and cold percentile temperature? Can you add a legend or describe it in the caption?

Thanks for your comment. We have now added a legend to identify the different lines and make the figure easier to understand.

Line 126: Is this because there is only one ensemble member per model for PDRMIP simulations? If so, it would be helpful to state this explicitly here.

Thanks for pointing this out. Yes, this is due to fact that only one ensemble member is available for the PDRM simulations. We added the following line (L126): The different definition of an extreme event compared to CMIP6 is due to the fact that PDRMIP only comprises one ensemble member per model.

Fig. 2: Is there a reason for highlighting these 3 models in particular? NorESM1 appears to be highlighted because the response to CO2 is the smallest but the reason for highlighting the other two models is not provided

We have now updated Figure 2. Now it shows the multi-model mean across both concentrationand emission-driven simulations.

Line 144: What is the significance level and is the statement here that the spatial patterns of changes in SD and number of extreme days significantly different or correlated? We have now added the significance levels to the text. We further modified the text to say (L???) "Standard deviation differences are significant at p-level ; 0.05 using the Kolmogorov–Smirnov test." to highlight that the SDs are statistically different.

Line 146-147: I'm confused by this interpretation. How does a higher correlation between changes in SD and extremes evaluated separately for different forcings tell us how the PDF change is more dependent on changes in aerosol vs CO2? Since SD is influenced by changes in both low and high precip extremes, my interpretation of this result is that the response to aerosol forcing disproportionately affects high vs low precip extremes and thus is better captured by SD, whereas the response to CO2 forcing may also be affecting low precip extremes

Thanks for your comment. We are not looking at whether aerosol forcing disproportionately affects high vs low precipitation extremes (note that the correlation values are spatial correlations between Figures A1-A3 and A4-A7). Here, we meant that aerosols lead to a wider/narrower distribution than  $CO_2$ . We have added the following text in Lines 163-165 for clarification:

"The results show a higher correlation between changes in SD and extremes for the aerosol simulations than for the  $CO2 \times 2$  experiment. This indicates that aerosols lead to a wider/narrower distribution and thus more days of extreme precipitation than the influence of  $CO_2$ . Additionally, the effect of aerosols is highly regionally dependent whereas the PDFs to a  $CO_2$  increase are getting wider over all regions."

*Fig. 3 caption: Do you mean p j 0.05?* This is fixed

Line 165: It looks like an increase in the number of intense precipitation events due to daily variability change is not seen everywhere in Asia. For example in MPI-ESM1-2-LR, there is a decrease along the 30th parallel north and for ACCESS-ESM1-5 and CanESM5, there is a weak response in the Middle East and Central Asia. I suggest you specify the subregions within Asia where the response is large

Thanks for the suggestion, we now separately describe changes in East and South Asia in line 165.

Fig. 4 and 5: Is there stippling anywhere in these plots? I don't see them even when I zoom into individual subpanels.

Thanks for pointing this out. We found a bug in the plotting script and have not added stippling to these plots.

Fig. 5: It would be easier to compare the role of mean vs variability change if the same colorbar settings were used for both Fig. 4 and 5. Currently the colorbar in Fig. 4 saturates on the positive end at 5 whereas the colorbar in Fig. 5 saturates at 1

Thanks for your suggestion, we have now changed the colorbar to have the same limits for Figure 4 and 5.

Line 175 and Fig. 6b: I'm surprised the change in mean dominates in Eastern Brazil, Southern Africa, and Northern Australia. Looking at Fig. 4 and 5 it appears to me that the brown is a darker shade in Fig. 4 compared to Fig. 5 for both MPI and CanESM. Can you provide plots with the variability and mean components side-by-side?

Given the amount of information available, we had to make a choice on what to show in individual figures. We find it a bit inconvenient to plot the variability and mean side by side for 3 models and 4 warming levels, which is why we split them up. We see the point of the reviewers, but would like to keep the original split for clarity.

Line 181: What does "rarely observed" mean? Is it the 0.999 quantile? We mean the 0.999 quantile which we have added to the text and figure 1.

Line 188-191: So the results shown here are not following the methods described in line 87-88? If so, you should mention in the methods section that you consider two different quantiles of extremes for CMIP6 data

We excuse the confusion: The CMIP6 results follow the method described in lines 87-88, while for the PDRMIP analysis a different quantile is used.

Line 201-202: Since not all of the domain shows a decrease in days of extreme precipitation I suggest you specify the region (e.g., South, Southeast, and East Asia)

We have modified the text and specified the regions as suggested. The CanESM5 model suggests that with a continuous reduction in aerosol emissions, an increase in the likelihood of extreme precipitation events is continuously reduced in South and East Asia in the near future.

**Line 203-204: What you are describing here is only apparent around the Tibetan Plateau. I suggest specifying this.**

We modified the text as follows (L220): "In contrast, MPI-ESM1-2-LR indicates a slight decrease in extreme weather events from 2025 to 2034, followed by an increase from 2035 to 2044 over Tibetian Plateau."

Line 205: What does significant mean here? Statistically significant? Significant for impacts? We changed the word "significant" to "prominent" to avoid confusion.

Line 222-234: Do you follow a systematic process for labeling a distribution as Gaussian, Gammalike, or exponential-like? Otherwise characterizing the distributions like this is subjective and arbitrary. What are the standard deviation, skewness, and kurtosis of these distributions

We added a figure showing the evolution of regional mean standard deviation, kurtosis and skewness, to better present the evolution of PDFs. We further changed the text (L239-242) to:

"For SAS, the underlying PDFs are quite different between the individual models. The most prominent difference relates to changes in the kurtosis (Figure D1). ACCESS-ESM1-5 and CanESM5 show higher kurtosis values than MPI-ESM1-2-LR. CanESM5 is the only model that shows decreasing kurtosis with global warming. Nonetheless, all models show a similar widening of the distributions with global warming.

Over the WAF region, all three models exhibit similar skewness evolution. With global warming, the MPI-ESM1-2-LR and CanESM5 distributions are getting wider, indicating an increase in daily variability and an associated increase in extremes at both ends. CanESM5's evolution in standard deviation plateaus after two degrees of global warming. While CanESM5 shows a widening of the PDF, similar to the other two models, it also shows a clear change in the mean of the distribution. As a result, the likelihood of extreme values is primarily increasing at the high end of the tail."

**Line 231: Have you quantitatively tested the similarity between different distributions? (e.g., Kolmogorov Smirnov test)**

We have performed a KS test for the distribution and found that the distributions are different. We have now modified the text as follows (L250):

"Over NWN, all three models exhibit similar PDF shapes (while their distributions are statistically different). However, the model responses diverge regarding the PDF evolution under global warming. CanESM5 shows a change in the mean and little change in the width, while ACCESS-ESM1-5 and MPI-ESM1-2-LR changes are mostly in width and shape. Despite these discrepancies in underlying PDFs, all three models show a robust increase in summertime variability under global warming, which leads to an increased likelihood of precipitation extremes in the Arctic, Asia and Africa."

**Line 235-239: Can you discuss the regional dependence of the relative role in SD vs skewness change? For example in MPI, SD change is particularly large in WAF. In CanESM, SD change is large in SAS**

We have added an additional paragraph (L. 260):

"Another interesting aspect is the regional dependence on the relative roles of changes in SD versus skewness. Each region and model behaves differently. Over WAF, MPI shows large changes in both skewness and standard deviation, whereas CanESM5 shows small changes in standard deviation over WAF. Interestingly, CanESM5 shows the largest change in standard deviation over SAS. These observations highlight that each region responds differently to global warming and that there is significant model uncertainty regarding how variability changes."

**1.3 Technical comments**

Line 27: "uncertainties"  $\rightarrow$  "uncertainties" Fixed Line 121: "asses"  $\rightarrow$  "assess" Fixed Line 144: "test test"  $\rightarrow$  "test" Fixed Line 160 and 175: "South Africa"  $\rightarrow$  "Southern Africa"; South Africa is a country whereas Southern Africa refers to the southern half of the African continent. I assume you mean the latter vs the former. Fixed Line 171: "northern America"  $\rightarrow$  "North America" Fixed Fig. 9 colorbar label: "extream"  $\rightarrow$  "extreme" fixed Line 257: "underplaying"  $\rightarrow$  do you mean "underlying"? Fixed Line 282: remove "net" Fixed

**2 Referee 2. comments**

**2.1 Specific comments**

L33-34: Could you clarify how Asian aerosol emissions affect Arctic temperatures and the Australian monsoon? Local aerosol emission could result in direct radiative forcing through advection and could occur teleconnection pattern through forced Rossby waves. It would be great to clarify this part to justify next sentence about the impact of aerosol emission on local and remote regions

We changed lines 33-34 to "Asian aerosol emissions have, for example, pronounced effects on Arctic temperatures due to change in energy transport and albedo feedback (Merikanto et al., 2021) and the Australian monsoon due to changes in teleconnection patterns (Fahrenbach et al., 2024)."

1) What is meant by "single-model initial-condition large ensembles" in this section?

2) From "the second step ", did you remove the annual cycle from your definition of 'variability'?

3) You defined an extreme event as one that exceeds the 0.999th quantile (99.9th percentile) in one instance and then used a different definition of extreme events as those that exceed 99th percentile elsewhere.

I recommend clarifying this part of the methodology for the readers. 2. L75-90: This part is a little bit confusing, making it difficult to understand the method. I have a few questions and suggestions for clarification:

1) What is meant by "single-model initial-condition large ensembles" in this section?

2) From "the second step", did you remove the annual cycle from your definition of 'variability'?

3) You defined an extreme event as one that exceeds the 0.999th quantile (99.9th percentile) in one instance and then used a different definition of extreme events as those that exceed 99th percentile elsewhere.

I recommend clarifying this part of the methodology for the readers.

1) Here, the single-model initial-condition large ensembles are a set of model runs where the model is run with the same forcing, and physical parametrization but slightly different initial states. For ACCESS-ESM1-5, the experiment is branched from a progressively advanced year of the piControl simulations. For CanESM5, ensemble members were generated by launching historical runs at 50-year intervals off the piControl simulation. To explain this in more detail, we have modified the text (L??) "In SMILEs each model is run multiple times with the same forcing and model configuration but different initial state."

2) Thanks for your suggestions, we added further clarification (L94): "These results in PDFs for each GWL which differ only by the influence of change in standard deviation, kurtosis and skewness".

3) The 90th percentile is used only for the PDRMIP analysis as the PDRMIP experiments only have one ensemble member per model.

Furthermore, the authors mentioned that "these changes correlate with changes in the SD.", providing only a range of correlation coefficients from 0.22 to 0.49. I am curious about which region's variability could be explained by changes in them SD. I recommend including a spatial map to show the correlation between changes in SD and variability 3. L130-138: The authors described the overall features of daily precipitation variability driven by anthropogenic forcings. However, they presented results from only three models and particularly highlight NorESM, which has the weakest response of variability to doubled CO2 levels. Is there a specific reason why the authors chose to display results from only these models?

Furthermore, the authors mentioned that "these changes correlate with changes in the SD.", providing only a range of correlation coefficients from 0.22 to 0.49. I am curious about which region's variability could be explained by changes in them SD. I recommend including a spatial map to show the correlation between changes in SD and variability

We modify Figure 2 to show multi-model means for emission and concentration-driven models separately and stippling indicates model agreement. We cannot provide spatial maps of SD and variability correlation in this study as changes in SD cannot be distinguished from changes in over-all variability.

4. L157–167: The authors highlighted the important role of precipitation variability in the frequency of extreme precipitation events. Do you have any thoughts on which timescale of variability (or which physical phenomena) is associated with this? Thanks for your comment. We performed a power spectral analysis for the selected regions which revealed that overall variability is increasing on all time scales (and also outside of the selected regions). While the models and regions show some different behaviour, there are still common features across regions and models. For example, all models and regions show prominent changes on annual and longer timescales, which indicates that events which do not occur annually change most prominently. This likely suggests a link to ENSO-like variability suggested by REF (https://journals.ametsoc.org/view/journals/clim/30/11/jcli-d-16-0541.1.xml?tabbody = fulltext - display).

This level of detail is unfortunately out of the scope of what we can investigate within the current study. Here we focus on daily to monthly variability by using seasonal daily data. As our results broadly show a linear behaviour with the global warming, this would suggest that thermodynamic changes are a driver of this variability change. We have included the following paragraph to the discussion section (L267):

"What physical mechanisms drive the changes in variability (including the change in standard deviation and skewness)? Zhang et al. (2021) performed a moisture budget analysis on a Parameter Perturbed Ensemble of the HadGEM3-GC3.05 model. This approach also samples uncertainty from the model uncertainty space, while SMILES only samples uncertainty from climate internal variability. Their moisture budget analysis reveals that changes in variability are driven by vertical moisture advection and thermodynamics. A similar conclusion is shown by zhang2024anthropogenic in the observed increase in precipitation variability in ERA5. On longer time scales there might be a link to ENSO variability suggested by kohyama2017nonlinear."

**5. L182–184: Is there a figure for this example about ACCESS-ESM1-5?**

Please see Figure A7 in the appendix showing the total change of the number of extreme heat days for each model and four warming levels.

6. L196–199: This part is difficult to understand, specifically how the authors extracted the effect of aerosols from the difference between SSP3-7.0 and SSP1-2.6. Could you explain more details in why this difference represents the effect of aerosols?

In the method section we describe the following:

"SSP1-2.6 includes a rapid reduction in global aerosol emissions until 2050, except for an increase over southern Africa due to rapid industrialization. The aerosol emissions in SSP2-4.5 and SSP5-8.5 show a similar, but weaker, pattern, with a decrease over the NH and increase in the Southern Hemisphere (SH) as well as a strong Asian aerosol dipole (i.e., a large increase over South Asia and large decrease over East Asia) until the 2040s (Wilcox et al., 2020b; Samset et al., 2019a). The main difference between SSP2-4.5and SSP5-8.5 lies in the black carbon (BC) emissions from South Asia which show an increase and decrease until the 2040s, respectively, as well as the aerosol emissions over South America related to different rates of deforestation (Lawrence et al., 2016). SSP3-7.0 also shows an NH decrease and SH increase in emissions. However, the sulfur dioxide (precursor of sulfate aerosols) emissions stay nearly constant over East Asia but increase over South Asia, with opposite changes in BC emissions (Wilcox et al., 2020b). The comparison of climate responses under these different SSPs (SSP1-2.6 and SSP3-7.0), thus, allows us to investigate the influence of anthropogenic aerosols on the PDF changes, as greenhouse gas emissions remain relatively constant in these SSPs, only aerosol emissions are decreasing in SSP1-2.6."

7. L209–216: Does this part mean that CanESM5 and ACCESS-ESM1-5 show more reasonable response of precipitation extreme frequency to aerosol emission change?

We do not think that this is the case since it is just one source of uncertainty.

8. L220: Could you explain why the authors chose these regions?

These AR6 regions show the most drastic change due to changes in variability. We added the

following sentence (L. 241): "All these regions show a significant increase in the number of intense precipitation days due to changes in variability."

9. L226 –230: ACCESS-ESM1-5 doesn't seem to be widen across different GWL.

Thanks for your comment, we have rewritten this section as follows (L249): Over the WAF region, all three models exhibit a similar evolution in skewness. With global warming, the MPI-ESM1-2-LR and CanESM5 distributions are getting wider, indicating an increase in daily variability and an associated increase in extremes at both ends. CanESM5's evolution in SD plateaus after two degrees of global warming. While CanESM5 shows a widening of the PDF, similar to the other two models, it also shows a clear change in the mean of the distribution. As a result, the likelihood of extreme values is primarily increasing at the high end of the tails.

**2.2 Technical corrections**

1. L14:2020-2040 à 2020-2040 2. L28, L29:Chen et al. (2021) -¿ (Chen et al., 2021) Samset (2022) -¿ (Samset, 2022) fixed

3. L31, L33: emissions(... -z emissions ( temperatures(... -z temperatures ( fixed

4. *L82:* ' - ¿ " fixed

5.  $L97:time period - \dot{\delta} time period$  fixed

6. L121:asses - i assess fixed

7. *L124:Typo? year2000* fixed

8. L144:test test -i test fixed

9. L191: B2, B3 -¿ B2, and B3 fixed

10. L237: CanEMS5 -
¿ CanESM fixed

**3 Referee 3. comments**

**3.1 Specific comments**

Section 2.1: Although the methodology follows Samset et al. (2019b), the methods section can benefit from a clearer explanation of the process.

Thanks for your suggestion, we have rewritten the method section to clarify the methodology and am now using more specific definitions for the PDFs.

Figure 1: Clarify the colour coding used in the figures, especially in Panel b of Figure 1, to avoid confusion with the colour scheme used in Panel a. Thanks for your comment, we have added a legend.

Line 88 and Figure 1: The text and the figure should be consistent regarding the threshold for defining extreme events. Given that the 0.999th quantile is mentioned in the text, the figure should be adjusted to reflect this if that is the correct threshold used in the study. We have updated Figure 1 accordingly.

Line 124: The text mentions using the multi-model mean across eight CMIP5-generation models for PDRMIP, but the table actually lists nine models. Fixed

Line 136: The text states: "Other common features among the models include a drying over the southern part of Europe." However, Figure 2 specifically shows "Changes in the average number of days per year of extreme (0.90 quantile) precipitation." The figure indicates a decrease in the number of days with extreme precipitation, not necessarily "drying," which could imply a general decrease in precipitation, not just extremes.

Thanks, we have updated the text as follows (L151): "Other common features among the models include a decreasing number of intense precipitation over the southern part of Europe."

Line 138 and Line 144: The text mentions changes in the standard deviation (SD) but does not reference the specific figures (Figure A4, A5, A6) that show these changes. The text should include references to these figures to guide the reader to the relevant visual data. We have now added the according figure references.

Figure 2: The caption of Figure 2 states: "Panel titles indicate if a model is emission- (emi) or concentration-driven (conc)." However, this information is not included in the figure itself. The figure should have this information clearly indicated in the panel titles. Although Figure A1, Figure A2 and Figure A3 with nine models are provided, it would be good to state why were those three models selected? HadGEM2 (emi) in Figure A3 certainly shows more changes than the selected HadGEM2 (conc).

We have updated Figure 2, which now shows the multi-model mean of both concentration- and emission-driven models.

Figure 3, Figure 4 and Figure 5: The text mentions stippling as a method to indicate regions where changes in the probability distribution functions (PDFs) are significant at  $p \ge 0.05$ . However, in Figures 3, 4, and 5, the stippling is either not visible or absent. The absence of stippling suggests that the figures do not highlight areas where the changes are statistically significant according to this criterion. This could be an oversight or intentional if no regions met the significance threshold, but without stippling, the figures do not convey this additional layer of statistical information. Thanks for your comment. We discovered a bug in the plotting script which we fixed now.

Figure 3 and Figure 4: Ensure the non-linear scale is intentional and justified, since the hue for -1 is the same as the hue for 5.

This is intentional since the smaller changes are very small but not equal to zero.

Line 160: The text should be revised to specify "northern hemisphere summer" or "boreal summer" when discussing seasonal changes in regions like the Amazon basin, South Africa, and Australia. This will ensure clarity and accuracy, as these regions do not experience "summer" in the same way as the Northern Hemisphere.

We changed the line to "However, there are notable exceptions: In regions like the Amazon basin,

Southern Africa, and Australia, there is a slight decrease in extreme precipitation events during the NH summer months."

Line 170: The use of the term "aridity" in the context of Figure 5 may not be appropriate. The figure illustrates changes in the number of extreme JJA precipitation events due to changes in the mean state under different global warming levels.

Thanks for spotting this, we have changed "aridity" to "dry days".

**3.2 Technical comments:**

Line 20: Cop,  $2023 \rightarrow$  Copernicus, 2023 (Please correct the formatting in References too). fixed

Line 28: Chen et al.  $(2021) \rightarrow (Chen et al., 2021)$  fixed

Line 29: Samset (2022)  $\rightarrow$  (Samset, 2022) fixed

Line 31: emissions(Persad,  $2023 \rightarrow \text{emissions}$  (Persad, 2023 fixed

Line 33: temperatures (Merikanto et al., 2021)  $\rightarrow$  temperatures (Merikanto et al., 2021) fixed

Line 82: "PDF of total change  $\checkmark \rightarrow$  "PDF of total change" fixed

Line 97: timeperiod  $\rightarrow$  time period fixed

Line 114:  $SSP2.4-5 \rightarrow SSP2-4.5$  fixed

Line 115: ACCESS-ESM5- 1  $\rightarrow$  ACCESS-ESM1.5 fixed

Line 121: asses  $\rightarrow$  assess fixed

Line 124: year2000  $\rightarrow$  year 2000 fixed

Line 144: test test  $\rightarrow$  test fixed

Line 153: (IPCC, 2021)  $\rightarrow$  IPCC (2021) fixed

Line 187: Above, we have shown, using idealized simulations performed as part of PDRMIP, the influence of different anthropogenic drivers on Earth's climate 3.1.  $\rightarrow$  Above, we have shown, using

idealized simulations performed as part of PDRMIP, the influence of different anthropogenic drivers on Earth's climate (Section 3.1). fixed

Figure 9 caption: extreme  $\rightarrow$  extreme fixes

Line 215: model (Ziehn et al., 2020  $\rightarrow$  model (Ziehn et al., 2020 fixed

---

## Referee Report (RR1)

The revisions the authors have made has improved the manuscript. However, there are some questions that remain unanswered and some new areas of confusion that have come up in the revised version.

Fig. 1: Can you describe what the vertical dashed lines mean in panels b and c?

Response to line 86 comment: "We do not further isolate the changes in SD and skewness. Figure 9 only shows how the SD and skewness change relative to each other."
I was referring to how the PDFs indicating standard deviation and skewness changes were calculated in Fig. 1c. Line 95-96 describes how the PDF associated with mean change is calculated but there is no explanation for how the SD change and skewness change is calculated in Fig. 1c. If the rest of the analysis does not isolate changes in standard deviation and skewness, why show PDFs that isolate their contribution here? I think this is an unnecessary source of confusion for readers. It seems that reviewer 2 also had a similar confusion about whether standard deviation is isolated.

On a related note, what is different about the analysis in Fig. D1 that allows standard deviation, skewness, and kurtosis to be isolated whereas they cannot be elsewhere? This needs to be clarified in the manuscript.

Response to Fig. 1b comment: "We have now added a legend to identify the different lines and make the figure easier to understand."
The legend is helpful. However, my second question about this figure remains unanswered. What is the location of this regional example? This needs to be specified for the results to be reproducible.

Line 122 - 123: "greenhouse gas emissions remain relatively constant in these SSPs [SSP1-2.6 and SSP3-7.0]"
I'm confused by the statement that greenhouse gas emissions remain relatively constant in SSP1-2.6 and more generally that the greenhouse gas emissions in these two scenarios are similar. Doesn't SSP1 involve a cut in greenhouse gas emissions? In terms of concentrations, this means the carbon dioxide concentration increases slowly and methane decreases by the middle of the 21st century whereas CO2 and CH4 continue to increase in the SSP3 scenario (see Figure 11 from Meinshausen et al. 2020, copied for convenience on the right). Are there studies that show that the difference between SSP3 and SSP1 is dominated by the aerosol effect compared to the greenhouse gas effect? If so, can you state this explicitly and provide references?

[Figure]

Line 160: "The spatial correlation between the SD and the number of extreme days."
Do you mean the spatial correlation between *changes* in SD and *changes* in the number of extreme days?

Line 161 - 162: "These SD differences are significant at a p-level < 0.05 using the Kolmogorov-Smirnov test."
Are you testing the null hypothesis that the change in SD is significantly different from 0? The current phrasing suggests this is what is being tested but this seems out of context from the previous sentence. Are you instead testing the null hypothesis that the change in SD and the change in the number of extreme days are correlated?

In either case I'm confused why the Kolmogorov-Smirnov test is used here. Are you comparing the change in PDFs or the spatial correlation between changes in standard deviation and changes in number of extreme days? If the significance of a correlation is being tested, shouldn't something like a Pearson's correlation test (or similar) be used instead of a Kolmogorov-Smirnov test?

Response to line 175 and Fig 6b comment: "We find it a bit inconvenient to plot the variability and mean side by side for 3 models and 4 warming levels, which is why we split them up."

Putting aside the decision to put these plots side by side, can you reconcile my original comment that the brown shades in Fig. 4 generally look darker compared to Fig. 5, yet Fig. 6 shows that changes in the mean dominate the change in extreme precipitation over parts of Eastern Brazil, Southern Africa, and Northern Australia? Is the result in Fig. 6 not intended to be consistent with a comparison of the colors in Fig. 4 and 5?

Response to line 231 comment: "We have performed a KS test for the distribution and found that the distributions are different."

I suggest you specify that you use the Kolmogorov-Smirnov test and the p-value of the test in the manuscript.

---

## Referee Report (RR2)

**Title:** Climate variability can outweigh the influence of climate mean changes for extreme precipitation under global warming

**General comments:**
The authors have made valuable improvements to their manuscript, although one response remains somewhat unclear. Overall, the revised manuscript is well-written and logically structured. It presents a novel perspective by suggesting a potential relationship between changes in aerosol emissions and the frequency of extreme summertime precipitation events. Pending minor revisions, I now believe this manuscript is suitable for acceptance.

**Minor comments:**
*3) You defined an extreme event as one that exceeds the 0.999$^{th}$ quantile (~99.9$^{th}$ percentile) in one instance and then used a different definition of extreme events as those that exceed 99$^{th}$ percentile elsewhere.*

*3) The 90th percentile is used only for the PDRMIP analysis as the PDRMIP experiments only have one ensemble member per model.*

I am curious about the authors' choice to use the 90th percentile as the threshold for defining an extreme event, specifically in the PDRMIP experiments, rather than applying the same criterion to the CMIP models. Could this decision be related to the relatively smaller number of extreme events in the PDRMIP experiments compared to the CMIP models?

---

## Author Response (AR2)

We thank the reviewers again for their helpful comments for improving our manuscript. The referee comments are shown with *blue font color and italics* , and our point-to-point responses with standard font.

**1   Referee 1. comments**

*Fig. 1: Can you describe what the vertical dashed lines mean in panels b and c? Response to line 86 comment: "We do not further isolate the changes in SD and skewness. Figure 9 only shows how the SD and skewness change relative to each other." I was referring to how the PDFs indicating standard deviation and skewness changes were calculated in Fig. 1c. Line 95-96 describes how the PDF associated with mean change is calculated but there is no explanation for how the SD change and skewness change is calculated in Fig. 1c. If the rest of the analysis does not isolate changes in standard deviation and skewness, why show PDFs that isolate their contribution here? I think this is an unnecessary source of confusion for readers. It seems that reviewer 2 also had a similar confusion about whether standard deviation is isolated. On a related note, what is different about the analysis in Fig. D1 that allows standard deviation, skewness, and kurtosis to be solated whereas they cannot be elsewhere? This needs to be clarified in the manuscript.*

Figure 1 is now change so that it includes only change in mean and variability. In figure text there is mention of model,and region used in panel b and also indicated what dashed lines mean. Figure D1 just shows calculated kurtosis,skewness and SD which are calcuatled from the underlayin PDF, not the effect of if only i.e kurtosis or skewness changees.

*Line 122 - 123: "greenhouse gas emissions remain relatively constant in these SSPs [SSP1-2.6 and SSP3-7.0]" I'm confused by the statement that greenhouse gas emissions remain relatively constant in SSP1-2.6 and more generally that the greenhouse gas emissions in these two scenarios are similar. Doesn't SSP1 involve a cut in greenhouse gas emissions? In terms of concentrations, this means the carbon dioxide concentration increases slowly and methane decreases by the middle of the 21st century whereas CO2 and CH4 continue to increase in the SSP3 scenario (see Figure 11 from Meinshausen et al. 2020, copied for convenience on the right). Are there studies that show that the difference between SSP3 and SSP1 is dominated by the aerosol effect compared to the greenhouse gas effect? If so, can you state this explicitly and provide references?*

To add more clarity we change the lines 122-123 to
    "The comparison of climate responses under SSP1-2.6 and SSP3-7.0, thus, allows us to investigate the influence of anthropogenic aerosols on the PDF changes as greenhouse gas emissions remain relatively constant in these SSPs and only aerosol emissions are decreasing in SSP1-2.6. We can estimate the effects of aerosols by comparing SSP1-2.6 with SSP3-7.0, as the most significant aerosol reductions occur in Southeast and South Asia under SSP1-2.6"
    We are not qunativying the excat impact of aerols, more like does they have effect at all on region variabilty on summer time precipiation

*Line 160: "The spatial correlation between the SD and the number of extreme days." Do you mean the spatial correlation between changes in SD and changes in the number of extreme day*
Yes, this is what we mean, this line (L160) is now corrected to "The spatial correlation between the change in SD and in the change of number of extreme days for SULx5"

*Line 161 - 162: "These SD differences are significant at a p-level ¡ 0.05 using the Kolmogorov-Smirnov test." Are you testing the null hypothesis that the change in SD is significantly different from 0? The current phrasing suggests this is what is being tested but this seems out of context from the previous sentence. Are you instead testing the null hypothesis that the change in SD and*

*the change in the number of extreme days are correlated?*

No, we are interested on regions where the underlying PDF is statistical different. To make this more cleared we added line L160. "Figure A5 shows spatial distribution of change in the PDF SD These SD differences are significant at a p-level ¡ 0.05 using the Kolmogorov–Smirnov test. The spatial correlation between the change in SD and in the change of number of extreme days for SUL×5 varies from 0.42 to 0.61 (Figure A5)"

Kolmogorov-Smirnov test is used because we are intrested are the samples drawn from same distribution or not.

*Response to line 175 and Fig 6b comment: "We find it a bit inconvenient to plot the variability and mean side by side for 3 models and 4 warming levels, which is why we split them up." Putting aside the decision to put these plots side by side, can you reconcile my original comment that the brown shades in Fig. 4 generally look darker compared to Fig. 5, yet Fig. 6 shows that changes in the mean dominate the change in extreme precipitation over parts of Eastern Brazil, Southern Africa, and Northern Australia? Is the result in Fig. 6 not intended to be consistent with a comparison of the colors in Fig. 4 and 5*

The figure 6 was generated using absolute values, which is now change to values actual change, and figures is drwan using change respect to 0 degree warming. We also removed line "In particular, changes in the mean state are the dominant driver of changes in extreme precipitation events over South America, Southern Africa, and Australia. Conversely, changes in variability play a more pronounced influence on extreme precipitation changes over Eurasia." from L194-195

*Response to line 231 comment: "We have performed a KS test for the distribution and found that the distributions are different." I suggest you specify that you use the Kolmogorov-Smirnov test and the p-value of the test in the manuscript*
We added line "To test if underlying PDF's are ly different we use Kolmogorov-Smirnov test and the p-value " to line 101

**2 Referee 2. comments**

*I am curious about the authors' choice to use the 90th percentile as the threshold for defining an extreme event, specifically in the PDRMIP experiments, rather than applying the same criterion to the CMIP models. Could this decision be related to the relatively smaller number of extreme events in the PDRMIP experiments compared to the CMIP models*
Yes,where with cmip6 runs we have multiple ensemble member, and with PDRMIP only 1 for 50 years, to get proper statistics of change in the extrems under different climate drivers we needed to use different criteria for extremes